



# Performance of reanalysis and mesoscale models for wind resource assessment off the coast of Hawaii

Lindsay M. Sheridan[1], Raghavendra Krishnamurthy[1], Tien Manh Nguyen[2], Yi-Leng Chen[2], William I. Gustafson Jr.[1], Ye Liu[1], Feng Hsiao[2], Rob K. Newsom[1], Preston Spicer[1], Evgueni Kassianov[1], Mikhail Pekour[1], Nicola Bodini[3], and Mark Severy[1]

[1]Pacific Northwest National Laboratory, Richland, Washington, United States
[2]University of Hawaii, Honolulu, Hawaii, United States
[3]National Renewable Energy Laboratory, Golden, Colorado, United States

*Correspondence to*: Lindsay M. Sheridan (lindsay.sheridan@pnnl.gov)

**Abstract.** The eastern Hawaii coast in the United States is characterized by considerable wind resource fuelled by persistent trade winds, making it an important area for wind energy research. The need is strong for reanalyses, higher-resolution regional simulations, and purpose-built wind datasets in locations where observations have been historically limited, such as Hawaii's offshore environments. However, studies using offshore observations in other parts of the world have shown that significant errors can occur in reanalyses and wind datasets, which can lead to inaccurate estimates of wind energy generation, payback periods, and extreme weather risks at project locations. The degree of such errors is influenced by a number of factors, including spatial resolution and the handling of processes within the planetary boundary layer (PBL). In this work, we provide a wind resource characterization from year-long lidar buoy measurements off the eastern coast of Oahu, Hawaii, an environment previously unobserved at the rotor level, and use it to establish the performance of four simulation datasets with distinct spatial resolutions and PBL representations. The Oahu deployment location is meteorologically unique and less complex compared with previous offshore wind performance study locations, being strongly characterized by the trade winds with minimal land-atmosphere interaction influences. Despite the unique and consistent conditions, we hypothesize that distinct simulation datasets will exhibit diverse ranges of errors similar to what has been seen for other offshore locations. We find the European Centre for Medium-Range Weather Forecasts (ECMWF) Reanalysis version 5 (ERA5) to strongly underestimate observed wind speeds at the Oahu location (bias = -1.53 m s$^{-1}$ at a hub height of 140 m), while a regional Weather Research and Forecasting (WRF) Model simulation produced by the University of Hawaii (UH-WRF) provides a significantly smaller wind speed bias (-0.25 m s$^{-1}$), highlighting the value of running regional, higher-resolution simulations. Despite not temporally overlapping with the Oahu deployment, the long-term annual average 140 m wind speeds from the 2023 National Offshore Wind dataset (NOW-23) and Global Wind Atlas version 3 (GWA3) produce smaller magnitude biases (+0.39 m s$^{-1}$ and -0.10 m s$^{-1}$, respectively) than ERA5. The large bias noted for ERA5 is driven by significant underestimation of fast wind speeds, which the study site is largely characterized by, along with discontinuities in the ERA5 diurnal cycle. We also speculate that the relative sparsity of observations for data assimilation in this remote part of the world could influence the performance of ERA5.





# 1 Introduction

Simulated wind data, including long-term reanalysis models and higher-resolution wind datasets, are essential for assessing
the offshore wind resource near Hawaii and locations around the world due to the scarcity of wind observational coverage
over the water. Many recent studies have assessed offshore wind energy potential using reanalysis models, in particular,
ERA5 (Hersbach et al., 2020) and the Modern-Era Retrospective analysis for Research and Applications version 2
(MERRA-2) (Gelaro et al., 2017), given their advantageous temporal and geographical coverages and their ease of data
accessibility versus observational campaigns in challenging marine environments (Soares et al., 2020; Hayes et al., 2021;
Nehzad et al., 2021; Soukissian et al., 2021; Cowin et al., 2023). At locations where offshore hub height wind measurements
do exist, wind researchers can follow the methodology of the land-based wind industry in using measure-correlate-predict
techniques, which are extensively reviewed by Carta et al. (2013), to extend the temporal coverage of the observations with
reference reanalysis data. Reanalysis models also serve as the forcings for higher-resolution datasets developed to serve the
wind energy community, such as Global Wind Atlas version 3 (GWA3) (Davis et al., 2023) and the 2023 National Offshore
Wind dataset (NOW-23) (Bodini et al., 2024a).

Despite the wide applicability and need for simulated wind datasets, such products contain inaccuracies with respect to
observations that can lead to significant errors in parameters relevant to wind energy researchers and developers, such as the
annual and long-term average wind speed, seasonal and diurnal trends in wind speed, and occurrences of weather-driven
events like wind ramps. Therefore, assessing and disseminating the performance of simulated datasets in previously
unstudied locations is necessary for understanding the risks of wind resource estimation and enabling the adjustment of
estimates as a means of improving accuracy. Due to the difficulty of collecting wind measurements over open water,
validations of reanalysis models and wind resource datasets are limited for offshore environments, particularly at heights
relevant to offshore wind turbine rotor layers, which on average covered heights between 32 m and 216 m for turbines
installed globally in 2023 (McCoy et al., 2024). However, floating lidar technology and tall meteorological towers deployed
on offshore platforms have increased opportunities to validate models and datasets in areas of offshore wind development
interest.

With the increasing availability (though still a relatively small sample) of offshore rotor level observations, studies have
emerged over recent years comparing the performance of multiple simulation datasets at turbine rotor heights in offshore
locations with the aims of aiding analysts in selecting the optimal datasets for resource assessments and highlighting areas
for accuracy improvement for dataset developers. These studies have shown significant differences in performance across
simulations being validated at a single offshore location, and in some cases one simulation product can be the best performer
for one error metric (bias, correlation, mean absolute error, etc) and the worst performer for another metric. For observed
wind profiles over the North Sea, Kalverla et al. (2020) found the Dutch Offshore Wind Atlas to have near zero bias, while
ERA5 underestimated the observed wind speed by approximately 0.5 m s$^{-1}$ regardless of height. At lidar buoy deployment
sites off the coasts of Virginia and New Jersey, United States, Sheridan et al. (2020) noted that ERA5, MERRA-2, the Rapid



Refresh (RAP) analysis, and the North American Regional Reanalysis tended to underestimate observed wind speeds, with MERRA-2 providing the smallest magnitude biases and RAP producing the largest magnitude biases at both sites when using a dynamic power law to adjust the model heights to the lidar height of 90 m. From a different lidar buoy off the coast of New Jersey, United States, Pronk et al. (2022) determined that ERA5 yielded a significant negative wind speed bias

(around -1 m s$^{-1}$), while the higher-resolution Wind Integration National Dataset Toolkit Long-term Ensemble Dataset (WTK-LED) had a reduced bias near -0.5 m s$^{-1}$. However, Pronk et al. (2022) found that ERA5 outperformed WTK-LED in terms of centered root-mean-square error (CRMSE) at the same site. Using observations from the same floating lidar as Pronk et al. (2022), Fragano and Colle (2025) established that NOW-23 significantly outperforms ERA5 in terms of wind speed bias, especially during the spring when they found biases ranging from -0.35 m s$^{-1}$ at 40 m to -0.90 m s$^{-1}$ at 140 m for

ERA5 and +0.25 m s$^{-1}$ at 40 m to -0.30 m s$^{-1}$ at 200 m for the regional offshore wind dataset NOW-23. Despite NOW-23 showing significantly lower wind speed biases than ERA5 off the coast of New Jersey, Fragano and Colle (2025) found that NOW-23 had notably larger wind speed mean absolute errors than ERA5 in all seasons except spring. Based on lidar buoy data off the California, United States coast, Sheridan et al. (2022) validated five models and found that MERRA-2 produced the smallest wind speed bias (near zero) at a site off northern California but the largest magnitude bias (-1.6 m s$^{-1}$) at a site

off central California, while RAP provided the highest correlations for both sites (0.88 and 0.94).

Variations across wind simulations for performance metrics like bias and correlation occur for a number of reasons. In their evaluation of reanalysis products, Ramon et al. (2019) found that the lowest correlations for wind speeds compared with global tall tower observations corresponded to the coarsest resolution grids. Similarly, Kalverla et al. (2020) attributed ERA5's horizontal resolution to underestimation of observed offshore wind ramps due to limitations in the model representation of the small-scale structures responsible for ramps. Sheridan et al. (2022) noted that the high correlation of

RAP with offshore California observations was at least partly due to model's higher resolution and therefore better ability at resolving coastal features and phenomena that coarser models miss. Pronk et al. (2022) determined that the preliminary WRF simulations for WTK-LED outperformed ERA5 in terms of bias at an onshore site and an offshore site in the United States but found the opposite trend for CRMSE. Pronk et al. (2022) suspected that the underperformance of WTK-LED for

CRMSE is due to WTK-LED's exaggeration of the diurnal cycle of wind speeds at both study sites, especially the onshore location. Bodini et al. (2024b) tested two PBL schemes in simulations off the coast of California using lidar buoy observations and established that using MYNN overestimated stability compared with observations and the YSU-based simulations, resulting in overestimation of offshore hub height wind speeds in the region using MYNN and the selection of YSU as the PBL scheme for the NOW-23 South Pacific region. Nunalee and Basu (2014) tested six PBL schemes within

WRF to replicate low-level jet (LLJ) events off the United States Northeastern Seaboard, which are impactful on the available wind resource for turbine production, and found that all of them underestimated jet core intensity and struggled to represent a weak LLJ but accurately simulated the timing and termination of jet development and termination. Nunalee and Basu (2014) noted inconsistencies among the PBL schemes for replicating LLJs, such as the Mellor-Yamada-Janjic and





Quasi Normal Scale Elimination parameterizations performing the worst and subsequently the best for two LLJ events

separated by 1.5 days.

To address the need for wind resource characterization and model validation in offshore environments, the U.S. Department of Energy (DOE) has collaborated with the Bureau of Ocean Energy Management (BOEM) to deploy multiple buoy-mounted research lidars in locations that hitherto have not been explored in an observational manner at relevant turbine heights. In late 2022, one of the lidar buoys was deployed off the eastern shore of Oahu to gather metocean observations in

support of future offshore wind development. While the first offshore wind call areas in Hawaii were located northwest and southwest of Oahu (McCoy et al., 2024), the eastern side of the island has optimal exposure to the northeast trade winds that dominate the region and the deployment offers the first observation-based understanding of the rotor-level wind resource in the area. The Oahu lidar buoy deployment lasted a period of one year, which captured a full seasonal cycle of offshore wind observations. In addition to the onboard lidar, the buoy was equipped with a suite of surface meteorological and oceanic

instruments to produce a more complete analysis of metocean impacts on the wind resource.

With the knowledge in mind that no simulated dataset will perfectly replicate wind observations, we conjecture that some datasets will perform better than others in representing the wind resource at the previously unstudied buoy deployment location off Oahu, leading to recommended use cases of these products for analysts in the wind energy community. In particular, we hypothesize that ERA5 will exhibit a low wind speed bias at our study site akin to previous offshore

evaluations and that higher-resolution datasets will find greater success at representing the observations. While many of the previous offshore wind validations have occurred in locations regularly influenced by nearby land-atmosphere interactions, like sea breezes and low-level jets, this study presents a unique look into model performance at the rotor level in a trade wind dominant environment. Our analysis evaluates the successes and challenges of four diverse datasets, ranging from a geographically coarse reanalysis product (ERA5) to a high-resolution climatology dataset (GWA3), in representing the

observed winds from the Hawaii lidar buoy deployment. To set the stage for the validation, we begin in Section 2 with a discussion of the observed meteorological findings from the lidar buoy and then describe each wind dataset and the methods for evaluation. Section 3 provides the assessment of the two datasets that temporally overlap with the lidar buoy deployment: the reanalysis ERA5 and a regional WRF simulation produced by the University of Hawaii, UH-WRF. Due to the concurrency of the observations and estimates, this section goes into detail on the dataset performance according to factors

like stability, wind speed class, and various meteorological phenomena that occurred during the deployment. Section 4 explores the similarities and differences between the lidar buoy observations and the two datasets that are not concurrent: NOW-23 and GWA3. We keep this evaluation a high-level comparison to determine whether the observed wind speeds and trends are represented in the ranges of simulated long-term wind speeds, as we acknowledge that the significant limitation of temporally misaligned datasets does not allow for a comprehensive validation. Finally, Section 5 ties the results from the

wind speed evaluations to impacts on wind energy estimation and concludes with recommendations for improving offshore wind resource assessment based on this and other analyses. The manuscript also provides Appendix A, which reports on the recovery, quality, and processing of the data from the Hawaii deployment.

## 2 Data and methodology

DOE owns multiple AXYS WindSentinel™ buoys, including the buoy deployed for resource assessment off the coast of
Oahu which is outfitted with a Leosphere Windcube v2 lidar system and surface meteorological and oceanographic
instruments (Figure 1). Prior to deployment off of Oahu, the lidar buoy underwent validation at Woods Hole Oceanographic
Institution's Martha's Vineyard Coastal Observatory from January to June 2020 and was subsequently deployed off the
Humboldt County, California coast from October 2020 to December 2021. The validation at Martha's Vineyard utilised an
International Electrotechnical Commission-certified reference lidar atop an offshore platform approximately 250 m from the
lidar buoy. The validation produced wind speed coefficients of determination ($R^2$) exceeding 0.98 and wind direction $R^2$
values exceeding 0.97 up to 200 m above sea level (a.s.l.) (Gorton and Shaw, 2020).

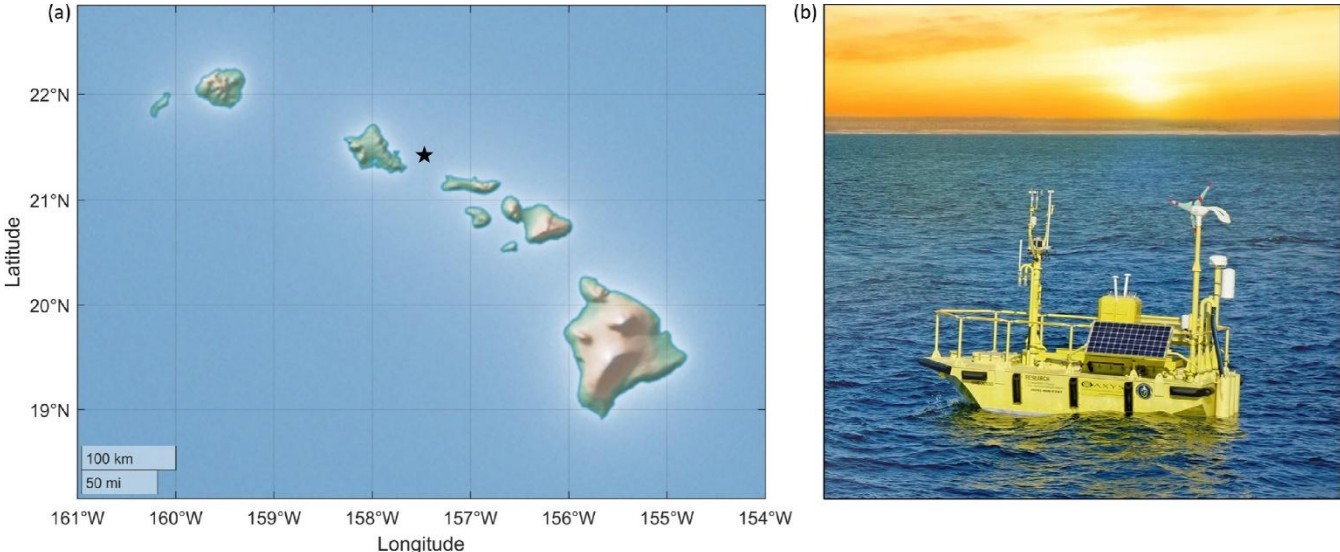

**Figure 1.** (a) Map of location of the Hawaii DOE lidar buoy deployment (indicated by the star). (b) Photo of one of the DOE lidar buoys
by Ocean Tech Services, LLC and Pacific Northwest National Laboratory.

### 2.1 Buoy observations

While the DOE buoy observations during the Hawaii deployment are discussed extensively in Appendix A, we provide the
key characteristics of the lidar wind measurements to set a baseline for comparison with the wind datasets. Many of the
observed characteristics are also found throughout the results to provide context for the evaluations of wind dataset
performance. The DOE buoy was deployed approximately 25 km off the coast of Oahu between 1 December 2022 and 15
December 2023 (DOE, 2025). The lidar aboard the buoy outputs wind data every 20 m between 40 m and 240 m. However,
early in the deployment it was noted that the 40 m wind output appeared corrupt, therefore this range gate was adjusted
higher in the atmosphere on 12 May 2023 with a new output height of 49 m. Following the quality control performed as





discussed in Appendix A, lidar data recovery for the deployment period is 98% or higher for all heights between 60 m and
240 m and greater than 99% for data at 49 m between 12 May 2023 and 15 December 2023.

For the buoy deployment period, annual average wind speeds range from 8.92 m s⁻¹ at 60 m to 9.41 m s⁻¹ at 240 m with very little shear across the profile (Figure 2a). The observed winds predominantly source from the east-northeast with very little veer across the profile (Figure 2b). Conversely to seasonal wind patterns determined over a long period of time, which tend to smoothly transition from one month to the next, the average wind speeds during the year-long Hawaii buoy
deployment are extremely inconsistent from one month to the next in the winter and fall (Figure 5). Using the 140 m output height as an example based on announced future offshore wind hub heights (McCoy et al., 2024), the monthly average wind speeds for the first four months of the deployment (December 2022 – March 2023) are consecutively 8.60 m s⁻¹, 6.30 m s⁻¹, 11.96 m s⁻¹, and 8.23 m s⁻¹. Following steadier transitions in the monthly average wind speeds throughout the spring and summer, the deployment concludes with another highly oscillatory period between September and November 2023 when the
monthly average 140 m wind speeds are consecutively 9.91 m s⁻¹, 5.97 m s⁻¹, and 9.78 m s⁻¹. The standard deviation of the 140 m monthly average wind speeds over the Hawaii deployment is 1.69 m s⁻¹. Conversely, the observed wind speeds transition steadily across the diurnal cycle, with the fastest wind speeds occurring during the evening and at night and the slowest wind speeds occurring during the day (Figure 6). The standard deviation of the 140 m hourly average wind speeds is 0.27 m s⁻¹.



**Figure 2.** Observed (a) wind speed and (b) wind direction by height a.s.l. from the Hawaii lidar buoy during 1 December 2022 – 15 December 2023.

## 2.2 Models and datasets

Reanalysis models support wind energy analysis in a variety of ways, including providing wind resource assessments and serving as the boundary conditions for higher-resolution modelling efforts. The global geographic coverage and long-term continuously updating temporal coverage of ERA5 allows for widespread use and validation (Table 1). ERA5 is developed by EMCWF and the hourly data on the lowest 9 of 137 model levels are used in this study (CDS, 2025). Data assimilation is performed using a 12-hour 4D-Var ensemble (Hersbach et al., 2020).

The University of Hawaii produced a year-long (2023) WRF v4.6.0 simulation (UH-WRF) with the innermost domain covering the islands of Maui, Lanai, Molokai, and Oahu (Table 1). The key configurations include the WRF Single-Moment 6-class Microphysics scheme for representing the cloud microphysical processes, the Betts-Miller-Janjic cumulus parameterization for convective processes, the Rapid Radiative Transfer Model for Global Circulation Models for longwave





and shortwave radiation, the Noah land-surface model, and the Yonsei University (YSU) planetary boundary layer (PBL) scheme (Hsiao et al., 2021). The lowest 6 of 51 model levels from UH-WRF are evaluated in this study at hourly resolution.

UH-WRF was initialized daily using ERA5 data beginning at 0 UTC each day with simulation hours t = 0-35. The land surface data, including terrain, soil type, ground vegetation cover, and soil moisture were updated following the procedures described by Zhang et al. (2005) and Hsiao et al. (2020). The model spin up time occurred between t = 0-11. For this analysis, we build the UH-WRF timeseries using forecast hours t = 12-35.

In addition to ERA5 and UH-WRF, which provide wind information along with a variety of other earth and atmospheric
variables, specific datasets have been developed for wind energy research and development. NOW-23, developed by the National Renewable Energy Laboratory (NREL), the University of Colorado, and Veer Renewables to support offshore wind research, consists of eight regional WRF model simulations: three for the Atlantic Ocean, two for the Pacific Ocean, one for the Great Lakes, one for the Gulf region, and one for Hawaii (Bodini et al., 2024a). The model parameters are custom to each regional simulation to account for geographically unique wind resource phenomena. The NOW-23 Hawaii simulation
uses ERA5 to provide the boundary conditions, the Mellor-Yamada-Nakanishi-Niino (MYNN) scheme for the PBL and surface layer schemes, the Noah land-surface model, and the Operational Sea Surface Temperature and Ice Analysis (OSTIA) for the sea surface temperature (Bodini et al., 2024a). Wind data are output at 18 heights between 10 m and 500 m at 5-min resolution (Table 1). Wind analysts can access NOW-23 and other wind datasets through the Wind Resource Database (NREL, 2025).

GWA3, developed by the Technical University of Denmark (DTU) and World Bank Group, utilises the WRF model with the Mellor-Yamada-Janjić (MYJ) PBL scheme and ERA5 as input and boundary conditions to produce simulated wind data with a 3-km horizontal resolution (Davis et al., 2023). From there, microscale modelling using the Wind Atlas Analysis and Application Program (WAsP) model is performed to achieve an output grid spacing of 250 m. GWA3 provides global coverage for land-based wind estimates and offshore wind estimates within 200 km of shorelines. Wind data are output at
five heights between 10 m and 200 m at annual, monthly, and diurnal temporal resolutions (Table 1). Users can access GWA3 through its web application (DTU, 2025).

**Table 1.** Characteristics of wind assessment products evaluated in this analysis.

| Product | ERA5 | UH-WRF | NOW-23 | GWA3 |
|---|---|---|---|---|
| **Type** | Reanalysis | Meteorological dataset | Wind assessment dataset | Wind assessment dataset |
| **Developers** | ECMWF | University of Hawaii | NREL, University of Colorado, Veer Renewables | DTU, World Bank Group |
| **Temporal coverage** | 1950 – present | 2023 | 2000 – 2019[a] | 2008 – 2017 |
| **Temporal resolution** | 1-hr | 1-hr | 5-min | Annual, seasonal, diurnal |
| **Spatial coverage** | Global | Hawaii | U.S. marine regions | Global |
| **Spatial resolution** | 31-km | 1.5-km | 2-km | 0.25-km |



| PBL handling | Data assimilation | YSU scheme | MYNN scheme | MYJ scheme |
|---|---|---|---|---|
| **Wind output heights used in this study** | Lowest 9 model heights | Lowest 6 model heights | 10 m, every 20 m between 20 m and 240 m | 10 m, 50 m, 100 m, 150 m, 200 m |

[a] Depending on the region, the temporal extent varies between 2019 and 2022. For Hawaii, the temporal extent of NOW-23 is 2019.

## 2.3 Validation methodology

For the two models that temporally overlap with the Hawaii buoy deployment, ERA5 and UH-WRF, the bias (Eq. 1), correlation (Eq. 2), and CRMSE (Eq. 3) are determined for the $N$ timestamps that the observed wind speeds ($U_{obs}$) and simulated wind speeds ($U_{sim}$) are available. The variables $\overline{U_{obs}}$ and $\overline{U_{sim}}$ are the average observed and simulated wind speeds, respectively, over the period $N$. The wind speed bias provides a gauge of whether the models tend to overestimate (positive bias), underestimate (negative bias), or accurately represent (zero bias) the observed wind speeds. The Pearson correlation coefficient explains the degree to which the simulated and observed wind speeds are linearly related, with values near 1 indicative of a high degree of correlation. The CRMSE portrays the degree of variation in error between the simulated and observed wind speeds, with larger values indicating larger errors.

$$bias = \frac{1}{N}\sum_{i=1}^{N}(U_{sim,i} - U_{obs,i}) \tag{1}$$

$$correlation = \frac{\sum_{i=1}^{N}(U_{sim,i} - \overline{U_{sim}})(U_{obs,i} - \overline{U_{obs}})}{\sqrt{\sum_{i=1}^{N}(U_{sim,i} - \overline{U_{sim}})^2}\sqrt{\sum_{i=1}^{N}(U_{obs,i} - \overline{U_{obs}})^2}} \tag{2}$$

$$CRMSE = \sqrt{\frac{1}{N}\sum_{i=1}^{N}\left((U_{sim,i} - \overline{U_{sim}}) - (U_{obs,i} - \overline{U_{obs}})\right)^2} \tag{3}$$

Analysis of the performance of NOW-23 and GWA3 off the coast of Oahu is significantly limited by the temporal misalignment between the dataset coverage periods (2000 – 2019 and 2008 – 2017, respectively) and the lidar buoy observation period (1 December 2022 – 15 December 2023). Therefore, we utilise the entire temporal coverage periods of NOW-23 and GWA3 to establish whether the observed wind speeds and the observed seasonal and diurnal wind speed trends are represented in the ranges of the simulated long-term wind speeds.

## 3 Year-long model validation

The following studies compare the performance of ERA5 and UH-WRF in representing key characteristics of the offshore wind resource at the location of the Hawaii lidar buoy deployment. For ERA5 and UH-WRF, the model level wind speeds are adjusted to the observed near surface (4 m) and lidar output heights (every 20 m between 60 m and 240 m) z using the



power law (Eq. 5) with the wind shear exponent α (Eq. 4) calculated at every timestamp using the surrounding model heights $z_{lo}$ and $z_{hi}$ and the associated wind speeds $u_{lo}$ and $u_{hi}$. The performance analysis of ERA5 and UH-WRF considers the entire wind profile up to 240 m (the limit of the lidar observations) along with a focused investigation at a single height, 140 m a.s.l., based on the announced hub heights for upcoming offshore wind projects (McCoy et al., 2024). Comparisons are performed using only hours when both the buoy observations and model estimates are available.


$$\alpha = \frac{\ln(u_{hi}/u_{lo})}{\ln(z_{hi}/z_{lo})} \tag{4}$$

$$u = u_{lo}\left(\frac{z}{z_{lo}}\right)^{\alpha} \tag{5}$$

### 3.1 Annual, seasonal, and diurnal performance

During the year-long buoy deployment period, ERA5 strongly underestimates the observed wind speeds across the profile with average biases ranging from -1.53 m s$^{-1}$ to -1.62 m s$^{-1}$ (-1.53 m s$^{-1}$ at 140 m). UH-WRF underestimates the observed wind speeds across the profile as well, but to a lesser degree, with average biases ranging from -0.24 m s$^{-1}$ to -0.29 m s$^{-1}$ (-

0.25 m s$^{-1}$ at 140 m). The correlations for both ERA5 (0.89-0.90) and UH-WRF (0.84-0.85) suggest successful representation of the hourly fluctuations in the observed wind speeds (Figure 3). Previous DOE lidar buoy deployments off the northern and central coasts of California revealed similar correlations for ERA5 (0.88) (Sheridan et al., 2022). The CRMSEs during the Hawaii buoy deployment (ERA5 = 1.58-1.60 m s$^{-1}$, UH-WRF = 1.96-2.02 m s$^{-1}$) (Figure 3) are notably lower than the ERA5-based CRMSEs found during the California deployments, which ranged between 2.3 m s$^{-1}$ and 2.4 m s$^{-1}$.

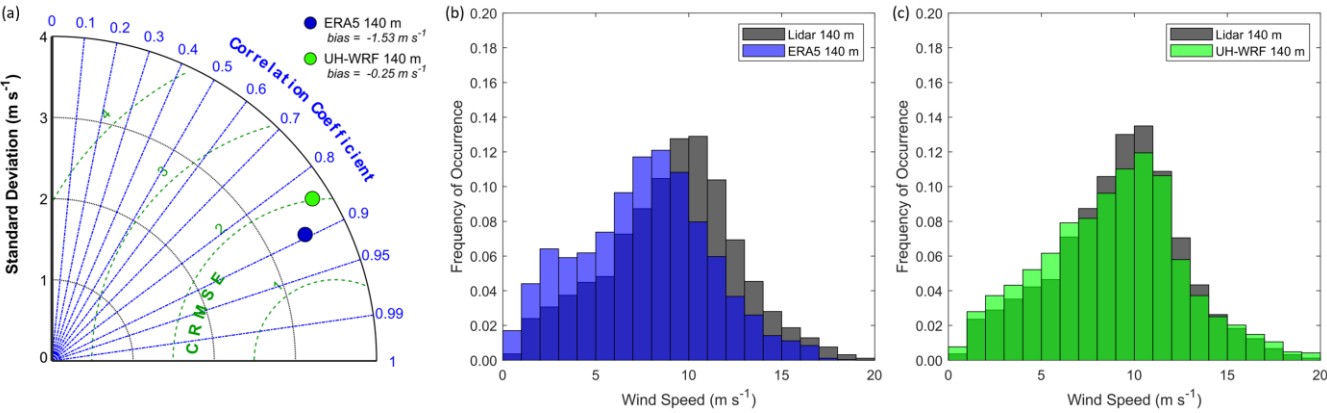


**Figure 3.** ERA5 (140 m, 1 December 2022 – 15 December 2023) and UH-WRF (140 m, 1 January 2023 – 15 December 2023) wind speed (a) correlations, standard deviations, and CRMSEs, and (b), (c), (d) distributions at the Hawaii buoy location.

The model wind speed biases during the Hawaii buoy deployment differ significantly depending on the magnitude of the observed wind speed. We use the NREL 15 MW offshore wind reference power curve (Musial et al., 2019) to categorize the



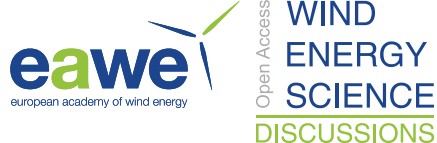

observed 140 m wind speeds according to pre-cut-in wind speeds ($\leq 3$ m s$^{-1}$), wind speeds along the steep portion of the power curve ($> 3$ m s$^{-1}$ and $< 11$ m s$^{-1}$), and wind speeds corresponding to maximum power production ($\geq 11$ m s$^{-1}$). During the full-year deployment period, 5.8%, 65.2%, and 29.0% of the lidar 140 m wind speeds fall into the pre-cut-in, steep portion, and maximum power categories, respectively. No observed or modelled wind speeds at any height between the surface and 240 m a.s.l. during the Hawaii buoy deployment exceeded the cut-out wind speed denoted by the reference

power curve (25 m s$^{-1}$), so biases during wind speed cut-out or derate periods are not able to be determined.

ERA5 and UH-WRF follow similar trends in bias according to the observed wind speed (Figure 4). In representing the observed pre-cut-in wind speeds at 140 m, ERA5 exhibits little bias (median = 0.09 m s$^{-1}$) while UH-WRF significantly overestimates (median bias = 0.83 m s$^{-1}$). For observed wind speeds on the steep portion of the power curve, ERA5 and UH-WRF underestimate with median biases of -1.49 m s$^{-1}$ and -0.25 m s$^{-1}$. The greatest ERA5 errors are determined for observed

wind speeds corresponding to maximum turbine generation, at the top of the power curve. For these faster observed wind speeds, ERA5 displays significant underestimation, with a median bias of -1.71 m s$^{-1}$, while UH-WRF produces a median bias of -0.30 m s$^{-1}$.

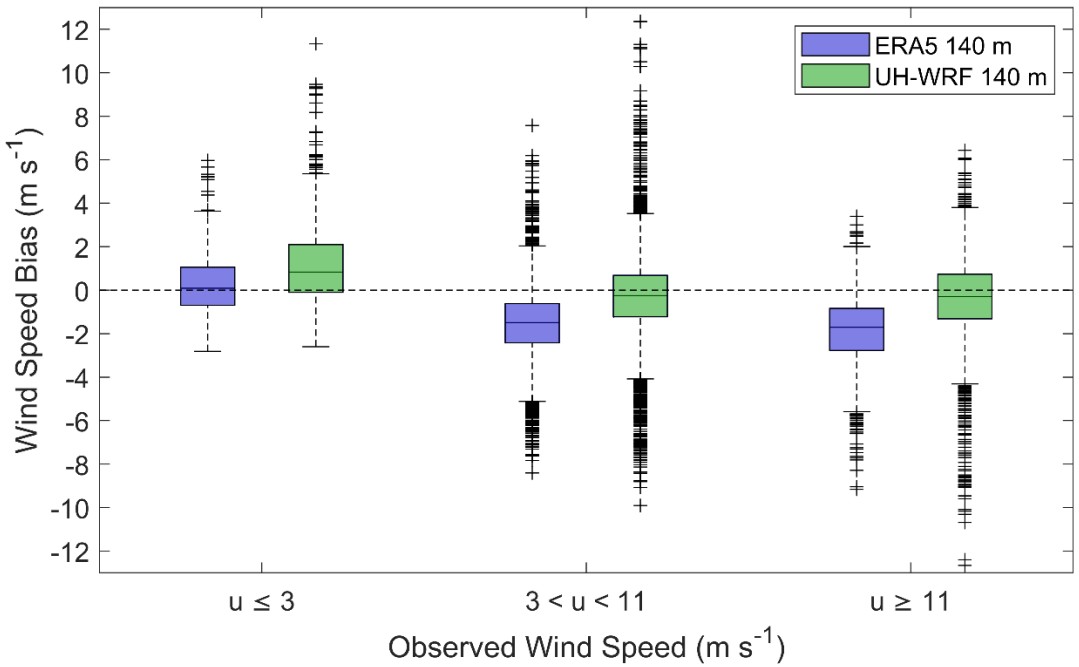

**Figure 4.** ERA5 and UH-WRF wind speed bias according to observed wind speed during the Hawaii buoy deployment.

The seasonal wind cycle during the lidar buoy deployment is characterized by immense variability in the fall and winter and more static conditions in the spring and summer (Figure 5). The observed monthly 140 m wind speed standard deviation is 2.30 m s$^{-1}$ for fall and winter months (September – February) and 0.79 m s$^{-1}$ for spring and summer months (March –



August). Especially notable is the steep observed monthly 100 m wind speed increase of 5.66 m s$^{-1}$ between January and February 2023 (Figure 5a).

Both ERA5 and UH-WRF perform well in capturing the observed monthly wind speed trends at the Hawaii buoy location. When comparing the observed and modelled normalized monthly wind speeds, ERA5 and UH-WRF produce correlations of 0.98 and 0.99, respectively (Figure 5b). The standard deviation of the observed normalized monthly wind speeds is 0.18 m s$^{-1}$, while the standard deviations from the models are slightly larger at 0.21 m s$^{-1}$ for ERA5 and 0.22 m s$^{-1}$ for UH-WRF.

The reanalysis bias varies throughout the year, with ERA5 exhibiting the largest biases in the spring (March – May 2023, average = -1.91 m s$^{-1}$), followed by the winter (December 2022 – February 2023, average = -1.50 m s$^{-1}$), the fall (September – November 2023, average = -1.43 m s$^{-1}$), and the smallest biases in the summer (June – August 2023, average = -1.34 m s$^{-1}$). UH-WRF follows a similar seasonal error pattern as ERA5, with the largest to smallest biases found in spring (-0.39 m s$^{-1}$), winter (-0.26 m s$^{-1}$ for January – February 2023), summer (-0.16 m s$^{-1}$), and fall (-0.13 m s$^{-1}$).

**Figure 5.** (a) Monthly average and (b) normalized monthly average wind speeds from the Hawaii lidar observations, ERA5 and UH-WRF. Average observed and modelled wind speeds for December 2023 reflect only the first half of the month, corresponding to the buoy deployment period.

The observed wind speeds transition gently across the diurnal cycle, with a standard deviation of the 140 m hourly average wind speeds of 0.27 m s$^{-1}$. The fastest speeds occur during the evening and at night, while the slowest speeds are present during the day (Figure 6a,c). The UH-WRF diurnal cycle follows that of the observations, with nearly zero bias occurring for hours between 0 and 11 UTC and underestimation occurring between 12 and 23 UTC (average hourly bias = -0.35 m s$^{-1}$). Contrary to ERA5's accurate representation of the observed seasonal wind speed cycle, the reanalysis struggles to capture the observed diurnal cycle. Trends in the ERA5 wind speeds and associated wind speed biases are tied to the start of the 12-hour reanalysis assimilation windows of 9 and 21 UTC (Hersbach et al., 2020) as shown by the sharp peaks at these hours in Figure 6. Such discontinuities in the ERA5 diurnal wind speed cycle are also noted by Kalverla et al. (2019)





over the North Sea. When comparing the observed and modelled normalized hourly wind speeds, ERA5 and UH-WRF produce correlations of 0.93 and 0.96, respectively (Figure 6c).

**Figure 6.** (a) Hourly average wind speeds, (b) hourly wind speed bias, and (c) normalized hourly average wind speeds from the Hawaii lidar observations, ERA5 and UH-WRF.



The wind resource in Hawaii is dominated by persistent northeast trade winds, with rare occurrences of wind sourcing from the south at Oahu (Argüeso and Businger, 2018). ERA5 underestimates the observed frequency of winds occurring

between 60° and 90° (60%) at 55% (Figure 7). During the overlapping period of the buoy observations and the UH-WRF simulation coverage, the percentage of winds occurring between 60° and 90° are very similar between the two datasets at 61% from UH-WRF versus 62% observed.

A small south-southwesterly component of the wind is identified in the lidar buoy observations, with 5% of the wind directions at 140 m occurring between 180° and 210°. While ERA5 captures a south-southwesterly wind component to

similar degree, at 4% of the observation period, the reanalysis slightly overrepresents the amount of wind sourcing from the southwest (between 210° and 240°), which includes the southern tip of Oahu (Figure 1a). ERA5 attributes the origin of 3% of the winds to the southwest as compared with less than 2% of the observed winds. During 2023, UH-WRF captures the south-southwesterly frequency (4% observed and simulated) and slightly overrepresents the southwest frequencies (2% versus 1% observed).

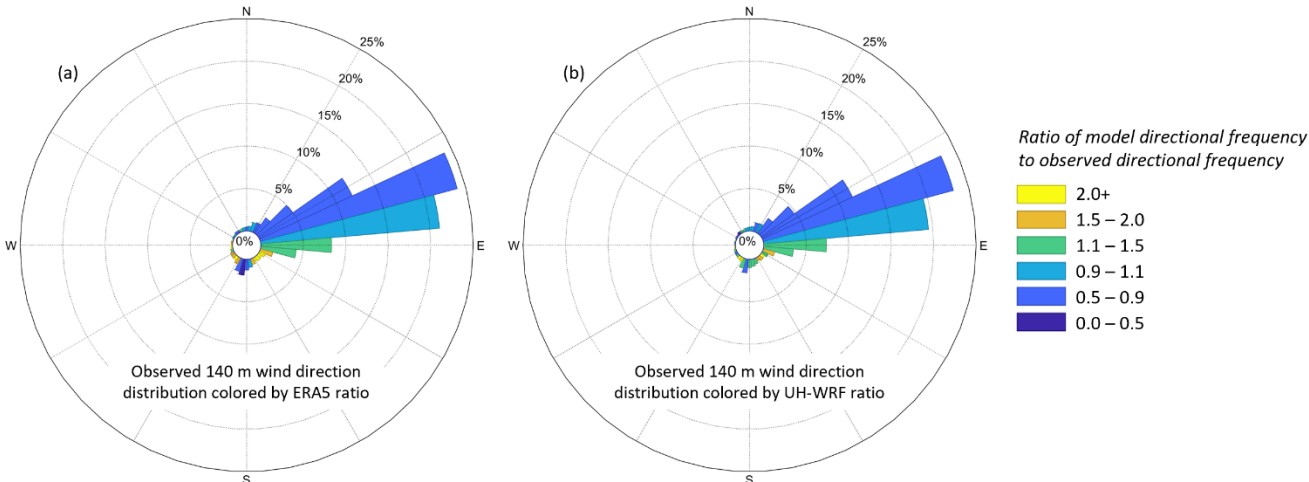


**Figure 7.** Observed wind direction distributions coloured by the ratios of (a) ERA5 and (b) UH-WRF directional frequencies to observed directional frequency by sector.

## 3.2 Performance according to atmospheric stability

The near surface atmospheric conditions over the Hawaii buoy deployment period are predominantly unstable, with 97% of

the period exhibiting cooler air temperatures and warmer sea surface temperatures (Figure 8a). No significant differences in the ERA5 and UH-WRF wind speed biases are noted for distinct atmospheric conditions. Underestimation of the observed wind speeds occurs during unstable and stable conditions, with median ERA5 wind speed biases near -1.50 m s$^{-1}$ regardless of whether the air-sea temperature differential is negative or positive (Figure 8b). For UH-WRF, the median wind speed biases are -0.27 m s$^{-1}$ for unstable conditions and -0.32 m s$^{-1}$ for stable conditions.





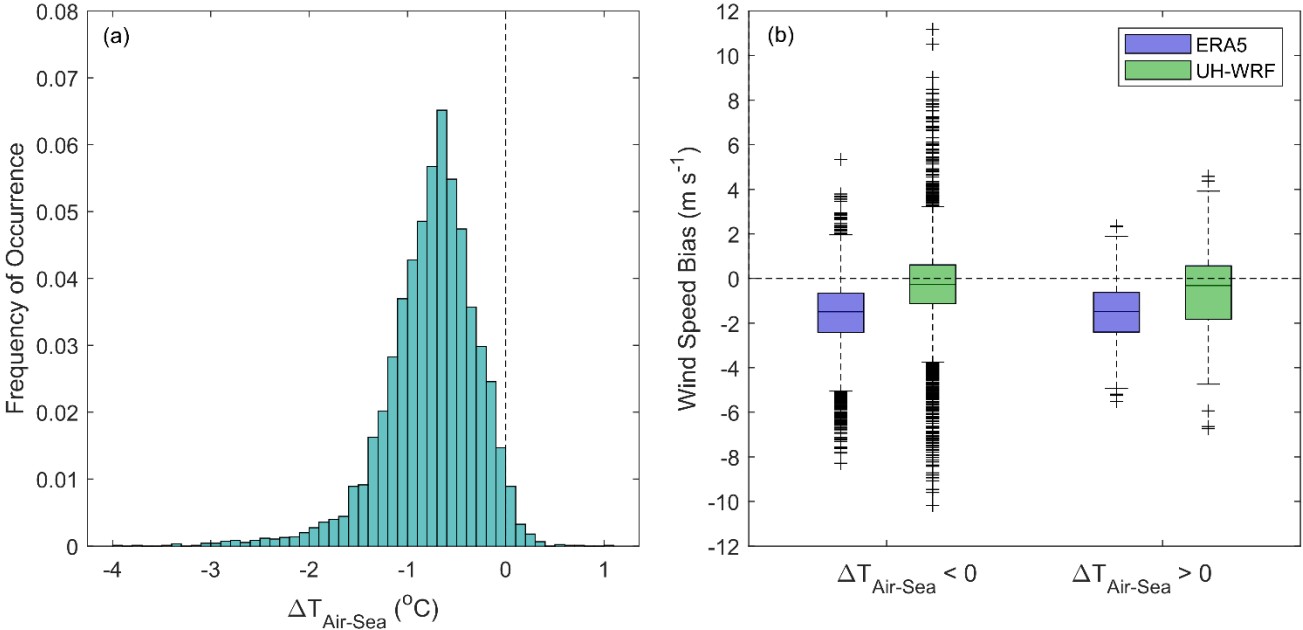


**Figure 8.** (a) Distribution of the air-sea temperature differential during the Hawaii lidar buoy deployment. (b) ERA5 and UH-WRF 140 m wind speed bias according to unstable ($\Delta T < 0°$) and stable conditions ($\Delta T > 0°$).

### 3.3 Wind shear analysis

The quantity of power that a wind turbine can produce is influenced by the amount of wind shear across the turbine rotor

plane (Wharton and Lundquist, 2012). The Hawaii buoy deployment period is characterized by very little shear across the wind profile. Using the wind shear exponent $\alpha$ (Eq. 4) calculated with the lidar wind speeds $u_{lo}$ and $u_{hi}$ at the output heights of $z_{lo} = 60$ m and $z_{hi} = 240$ m as our metric, we find that 78% of the wind shear exponents during the Hawaii deployment fall within $\pm 0.05$, compared with 43% of the central California and only 16% of the northern California deployments. ERA5 and UH-WRF, which similarly provide wind speed data at $z_{lo} = 60$ m and $z_{hi} = 240$ m, estimate even less wind shear across the

profile. ERA5 predicts 86% of the wind shear exponents during the Hawaii deployment to fall within $\pm 0.05$. During 2023, UH-WRF also predicts 88% of the wind shear exponents to fall within $\pm 0.05$, compared with 79% based on the lidar observations (Figure 9a,b). The ERA5 and UH-WRF wind speed biases tend to be negative during periods of observed negative and near-zero wind shear and become increasingly positive with larger shear exponents (Figure 9c).



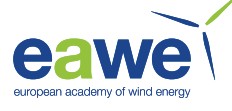 

**Figure 9.** Distribution of observed and (a) ERA5 and (b) UH-WRF wind shear between 60 m and 240 m during the Hawaii lidar buoy deployment. (c) ERA5 and UH-WRF wind speed bias according to observed wind shear exponent.

### 3.4 August 2023 Hawaii wind event

On 8 August 2023, stronger than normal northeast trade winds impacted the Hawaiian Islands and were a contributing factor in the fires that devastated the city of Lahaina on Maui (Mass and Ovens, 2024). On the same date, Hurricane Dora passed to the south of the islands (Bucci, 2024). To the east of the islands at the lidar buoy location, the observed and modelled winds shift further east-southeast from around 60° to 80° on 6 August (Figure 10b). Beginning on 7 August at 15 UTC, the buoy measurements capture a ramp event when the 140 m wind speed increases by 6.63 m s$^{-1}$ over the course of 5 hours (Figure 10a). ERA5 produces an increase in the 140 m wind speed beginning 1 hour later with a smaller magnitude ramp (4.42 m s$^{-1}$). The UH-WRF simulation shows an increase of 3.42 m s$^{-1}$ over 4 hours beginning at 15 UTC, followed by a stronger up





ramp a few hours later. ERA5 and UH-WRF have similar and lower correlations, respectively, during the event (0.89 and
0.81) versus the entire deployment (0.89 and 0.85). The ERA5 and UH-WRF wind speed biases at 140 m are smaller during
the event relative to the entire deployment (-1.36 m s$^{-1}$ versus -1.53 m s$^{-1}$ for ERA5 and 0.03 m s$^{-1}$ versus -0.25 m s$^{-1}$ for UH-
WRF).



**Figure 10.** Observed and modelled (a) wind speeds and (b) wind directions at the lidar buoy location during the August 2023 Hawaii wind
event.


## 4 Long-term model evaluation

This study proceeds to analyse the performance of two additional wind resource datasets off the coast of Oahu, NOW-23 and
GWA3, while acknowledging the significant limitation that these datasets do not temporally coincide with the lidar buoy





observation period (1 December 2022 – 15 December 2023). Therefore, we utilise the entire temporal coverage periods of NOW-23 (2000 – 2019) and GWA3 (2008 – 2017) to determine whether the observed wind speeds and the seasonal and diurnal wind speed trends are represented in the ranges of simulated long-term wind speeds. Given the higher temporal resolution of NOW-23, we additionally explore whether the occurrences of LLJs and ramp events are captured in the ranges of simulated long-term wind speeds for this dataset. The year-long period of 1 December 2022 – 30 November 2023 is used

for the observational point of comparison in the following analyses. As for the temporally concurrent assessments of ERA5 and UH-WRF, the analysis of NOW-23 and GWA3 considers the wind profile up to 240 m plus a focused investigation at 140 m.  GWA3 wind speed estimates are adjusted to 140 m following Eq. 4 and 5.

### 4.1 Annual, seasonal, and diurnal representation

Across the wind profile between 60 m and 240 m, the observed annual average wind speeds during the year-long Hawaii

deployment fall within the 25th and 75th percentiles of the annual average wind speeds simulated by NOW-23 and GWA3, though at distinct percentiles within that range (Figure 11). At 140 m a.s.l., the observed annual average wind speed during the period of December 2022 – November 2023 (9.12 m s$^{-1}$) is aligned with the 28th percentile annual average wind speed from NOW-23 over the full dataset period of 2000 – 2019. The median and average 140 m NOW-23 annual average wind speed estimates over the full dataset period are 9.43 m s$^{-1}$ and 9.52 m s$^{-1}$, respectively. The observed annual average wind

speed at 140 m coincides with the 57th percentile annual average wind speed for GWA3 during its coverage period of 2008 – 2017 (Figure 11b). The median and average 140 m GWA3 annual average wind speed estimates over the dataset coverage period are 9.06 m s$^{-1}$ and 9.02 m s$^{-1}$. Using the long-term annual average 140 m wind speed from NOW-23 produces a smaller magnitude bias (+0.39 m s$^{-1}$) than using ERA5 (-1.53 m s$^{-1}$), while GWA3 produces a smaller magnitude bias (-0.10 m s$^{-1}$)  than both ERA5 and UH-WRF (-0.25 m s$^{-1}$), despite the temporal concurrency with the observations of the latter two

products.







**Figure 11.** (a) NOW-23 (2000 – 2019) and (b) GWA3 (2008 – 2017) annual average wind speeds with observed annual average wind speeds (1 December 2022 – 30 November 2023).

The month-to-month variability in the observed 140 m wind speeds during the deployment year are not well captured by
the monthly wind speed indices provided by GWA3 or the median monthly 140 m wind speeds from NOW-23, with correlations of 0.40 and 0.29, respectively (Figure 12). GWA3 and NOW-23 are generally in agreement (correlation of 0.72 between the two datasets) that the slowest wind speeds east of Oahu occur in the winter and the fastest wind speeds occur in the summer, but the average observed winds during the year of deployment exhibit several significant inconsistencies between months, particularly during the periods December 2022 through April 2023 and September through November 2023
(Figure 12).

The range of the normalized observed wind speeds during the deployment year (0.65 in October 2023 to 1.31 in February 2023) exceeds those from GWA3 (0.83 in January to 1.10 in June) and the median of the 20-year NOW-23 dataset (0.89 in January to 1.11 in July). However, when considering all 20 years in the NOW-23 Hawaii dataset, normalized wind speeds at the location of the DOE buoy are estimated to reach extents of 0.50 and 1.45, with the upper estimate occurring the same

month as the fastest observed wind speeds, February (Figure 12). Performing a Student's t-test with a 5% significance level finds that December 2022, March and April 2023, June and July 2023, and November 2023 are within the variability of the NOW-23 climatologies.

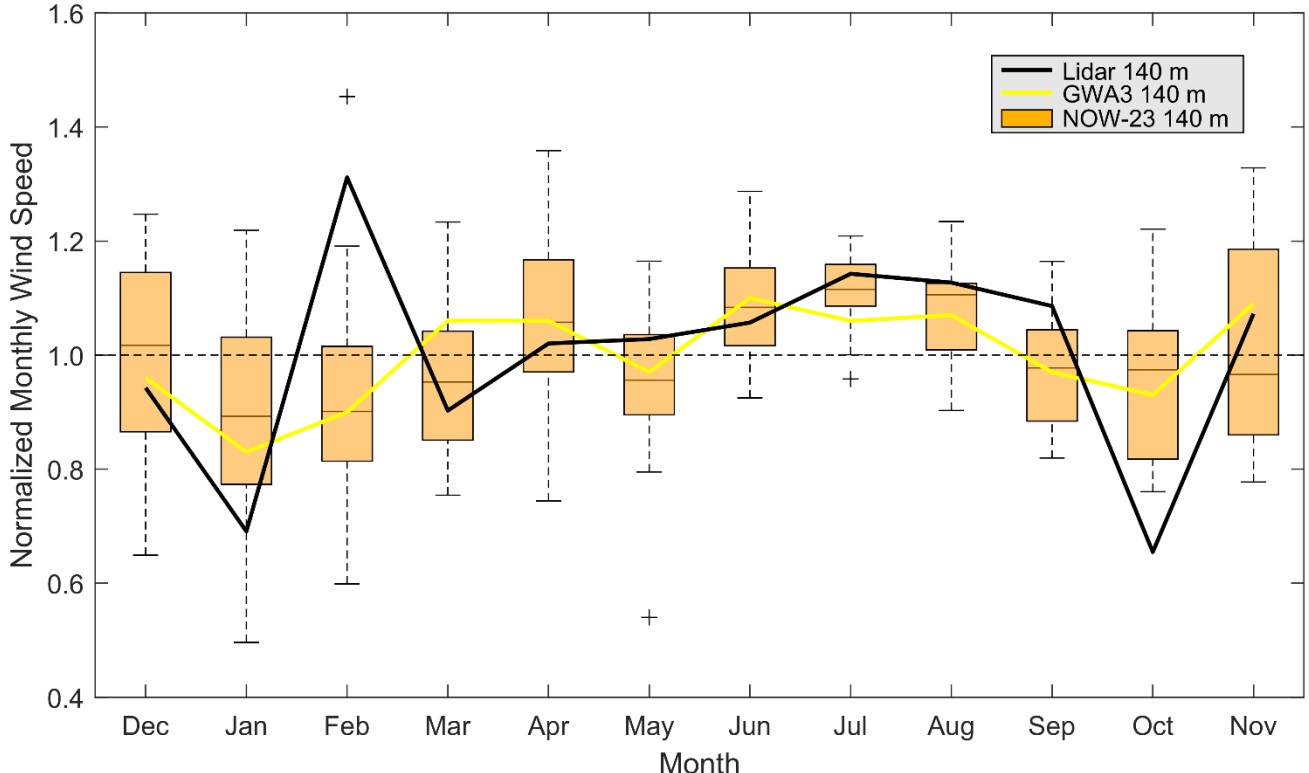

**Figure 12.** Normalized monthly wind speeds from the Hawaii deployment lidar (December 2022 – November 2023), GWA3 (2008 –
2017), and NOW-23 (2000 – 2019).

Conversely to the misalignment between the observed and simulated seasonal trends in the 140 m wind speed at the Hawaii buoy deployment location, the observations, NOW-23, and GWA3 agree that the fastest wind speeds occur during the evening and night and the slowest wind speeds occur during the daytime (Figure 13). The hourly wind speeds from NOW-23 and GWA3 achieve correlations of 0.94 and 0.89 with the observed hourly wind speeds, respectively. The range of

normalized wind speeds from the observations (0.96 to 1.05) is slightly greater than that of GWA3 and the median of the 20-year NOW-23 period (0.97 to 1.04 for both datasets). Disagreement between the observations and the datasets is greatest





between the hours 14 and 17 UTC, the morning transition when the air-sea temperature differential is at its greatest magnitude and begins approaching its smallest magnitude (Figure 13b).


**Figure 13.** (a) Normalized hourly wind speeds from the Hawaii deployment lidar (December 2022 – November 2023), GWA3 (2008 – 2017), and NOW-23 (2000 – 2019). (b) Diurnal distribution of the air-sea temperature differential during December 2022 – November 2023.




## 4.2 NOW-23 representation of ramp events

Observed wind ramp events are present at the Hawaii buoy deployment location, though not plentiful. Following the study of Bianco et al. (2025), we determine the amplitude of the change in the wind energy capacity factor, $\Delta CF$, using the rotor equivalent wind speeds as defined by Wagner et al. (2014) from the lidar and NOW-23 with the NREL 15 MW offshore wind reference power curve (Musial et al., 2019), over a duration $\Delta t$. For this study, we consider $\Delta CF \geq 0.4$ over $\Delta t = 10$ minutes through the analysis period December 2022 – November 2023. The year-long observations produce slightly more up

ramps (56) than down ramps (42), for a total of 98 ramps. The annual ramp estimates from NOW-23 are very similar to the observed at a median of 102 total ramps, though the median annual amount of up (down) ramps is slightly higher (lower) at 64 (38) (Figure 14).

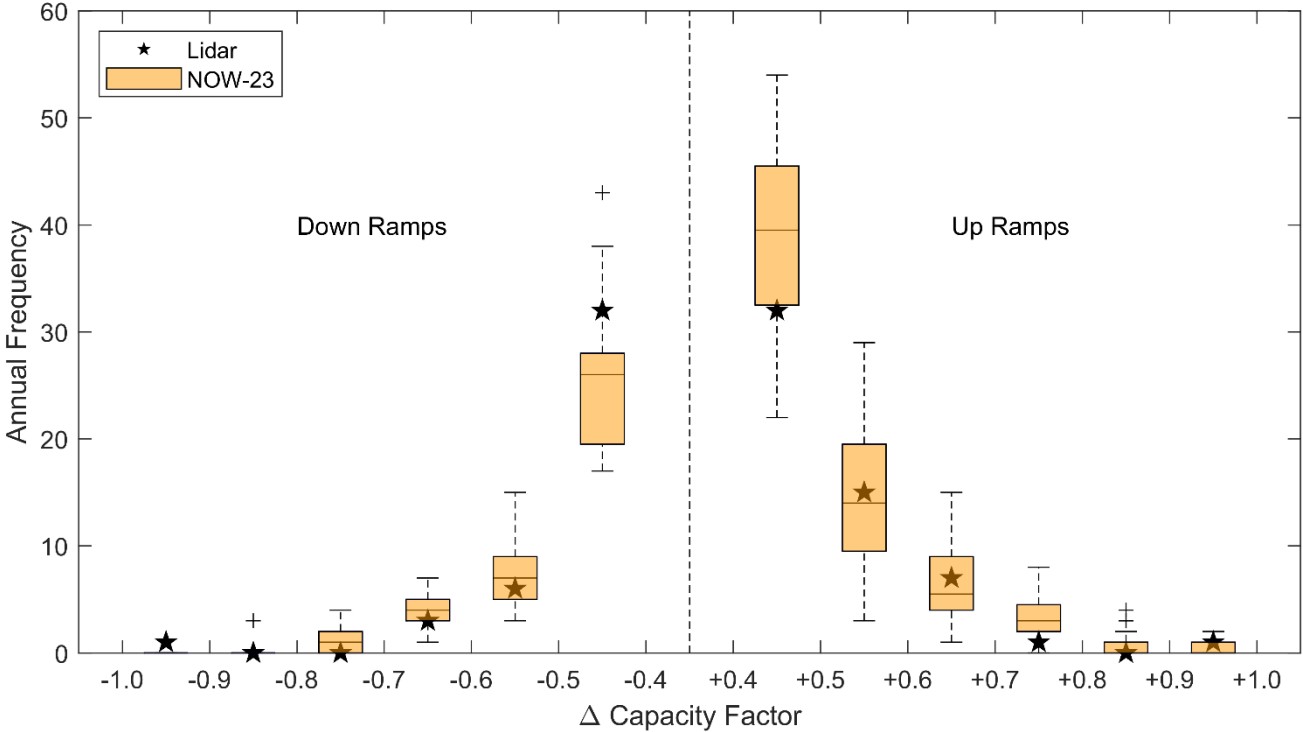

**Figure 14.** Annual frequency of observed and NOW-23 100 m wind ramps defined by a change in capacity factors of at least ±0.4 over 10
minutes.

## 5 Conclusions and implications for wind energy

In this study, we questioned and established the accuracy of reanalysis models, regional models, and purpose-built wind datasets in representation of the observed offshore wind resource in a previously unstudied location off the coast of Oahu, Hawaii. At a location of future offshore wind development interest due to vast wind resource availability and potential

economic impact, the year-long DOE lidar buoy deployment off the coast of Oahu highlights the successes and challenges of



a widely employed reanalysis and more recent datasets developed to support wind resource characterization. The wind speed biases quantified in this validation provide important baselines for researchers and energy developers deciding which models and datasets to use and how much to adjust their expectations when estimating the long-term offshore wind resource in the Hawaiian region. Given the consistency of the trade wind trends east of Hawaii, we recommend that offshore wind farm
planners in this area leverage the datasets found to be successful in this work and adjust their expectations when using datasets that did not perform as well according to their attributes of interest, such as wind speed bias and seasonal and diurnal wind variations.

In particular, the results of this investigation near Hawaii add another geographic data point to the trend of ERA5 underestimation of observed offshore wind speeds as documented by Kalverla et al. (2020) over the North Sea, Sheridan et
al. (2020) off the coasts of New Jersey and Virginia, United States, Pronk et al. (2022) and Fragano and Colle (2025) off the coast of New Jersey, United States, and Sheridan et al. (2022) off the coast of central California, United States. Given the consistent trade winds at the Oahu deployment site, we do not attribute the large ERA5 wind speed bias (-1.53 m s$^{-1}$) to the coarseness of the grid leading to mischaracterization of meteorological phenomena such as wind ramps or LLJs, simply because the observations yielded so few of these events, though such errors have shown to be significant at other offshore
sites where they are more prevalent (Kalverla et al., 2020; Sheridan et al., 2024). Similarly, we do not identify any performance trends according to atmospheric stability (Figure 8) such as are seen in the work of Pronk et al. (2022), who note the best ERA5 performance across multiple metrics occurring for unstable cases, followed by neutral conditions, and last, stable cases at their offshore study location. Rather, we note that ERA5 significantly underestimates fast wind speeds (Figure 4), which our study site is strongly characterized by (Figure 2). Gandoin and Garza (2024) provide evidence of
ERA5's underestimation of fast wind speeds and link the trend to ERA5's representation of the Charnock parameter. We also speculate that the relative sparsity of observations for data assimilation in this remote part of the world could influence the ERA5 error metrics, along with the discontinuities in the diurnal cycle noted in this work (Figure 6) and that of Kalverla et al. (2019). We recommend that users of ERA5 in the vicinity of the Oahu buoy deployment adjust the initial wind speed and energy generation expectations higher through bias correction (Wilczak et al., 2024) and welcome the use of our
publicly available buoy observations (DOE, 2025).

When financial and computational resources are available, the validation of the UH-WRF simulation in this work showcases the value of running higher-resolution regional simulations to improve accuracy in wind resource assessments. UH-WRF exhibits a significantly smaller bias (-0.25 m s$^{-1}$) than ERA5 and a correlation (0.85) not far behind that of ERA5 (0.90) at the Oahu site, supporting the selection of YSU as the PBL scheme in an additional offshore location (Bodini et al.,
2024b). While both ERA5 and UH-WRF successfully capture the observed seasonal wind speed cycle at the deployment location, UH-WRF is superior in capturing the diurnal cycle (Figure 6). While such regional simulations as UH-WRF can be temporally limited due to computational expense, researchers have an opportunity to develop long-term estimates by bias-correcting datasets with longer durations with high quality datasets of shorter duration (Buster et al., 2024).



Finally, we found it interesting that two datasets that lacked temporal overlap with the Oahu buoy deployment, NOW-23
and GWA3, outperformed ERA5 in terms of bias when simply using their respective long-term annual average wind speeds.
However, challenges arise when taking the comparisons to a more granular level, for example, the monthly pattern in the
wind observations did not align with the long-term seasonal trends estimated by NOW-23 and GWA3. Additionally, users of
GWA3 should be aware that the diurnal wind speed trend is currently available only for 100 m a.s.l. Based on our
evaluations and previous offshore wind validations, we recommend performing wind resource assessments using multiple
datasets to garner a range of expectations rather than selecting just one product that may result in significant error.

To conclude our study and highlight the results in a wind energy context, we translate the wind speed errors for each
simulation product to energy errors to assess what their impacts could be for an operating offshore wind farm. Following the
form of Wagner et al. (2014), we calculate the rotor equivalent wind speeds for concurrent timestamps across the lidar
observations, ERA5, and UH-WRF, assuming a hub height of 140 m. Applying the NREL 15 MW offshore wind reference
power curve (Musial et al., 2019) to the lidar-based rotor equivalent wind speeds from the year-long Oahu deployment
produces an estimated gross capacity factor of 59.8%. To explore the impacts of the wind speed biases noted for ERA5 (-
1.53 m s$^{-1}$ to -1.62 m s$^{-1}$) and UH-WRF (-0.24 m s$^{-1}$ to -0.29 m s$^{-1}$) (Section 3.1) across the rotor layer, we apply the same
power curve to the ERA5 and UH-WRF rotor equivalent wind speeds, which over the same time period results in estimated
gross capacity factors of 42.2% and 56.0%, 18 and 4 percentage points lower than the estimate using the observations,
respectively. On average over their respective temporal coverage periods, GWA3 and NOW-23 produce estimated capacity
factors closer to the capacity factor based on observations than does ERA5: 52.7% (GWA3) and 63.5% (NOW-23) using the
15 MW reference power curve.

Onsite measurements spanning wind turbine rotor diameters are critical to understanding the accuracy of the models and
datasets that drive the offshore wind industry through resource assessment, wind trend characterization, and downscaling.
Studies like this one highlight the wealth of information that can be obtained for establishing observation-based wind
characterizations, simulation accuracies, and simulation correction possibilities and encourage further offshore wind
observational campaigns in unstudied locations.

**Appendix A: Summary of Hawaii buoy observations and post-processing**

**A1 Instrumentation**

Pacific Northwest National Laboratory (PNNL) operates lidar buoys that are owned by the U.S. Department of Energy to
collect offshore wind resource information in areas of potential commercial development. The buoys have previously
collected measurements off the coast of Virginia, New Jersey, and California (Shaw et al., 2020; Krishnamurthy et al, 2023).
The buoy (Figure A1) is equipped with a wind-profiling lidar capable of measuring wind speed up to 250 m above the
instrument, surface meteorological measurements, oceanographic measurements, wave spectrum, and ocean current profile
were collected from the instruments installed on the buoy (Table A1).





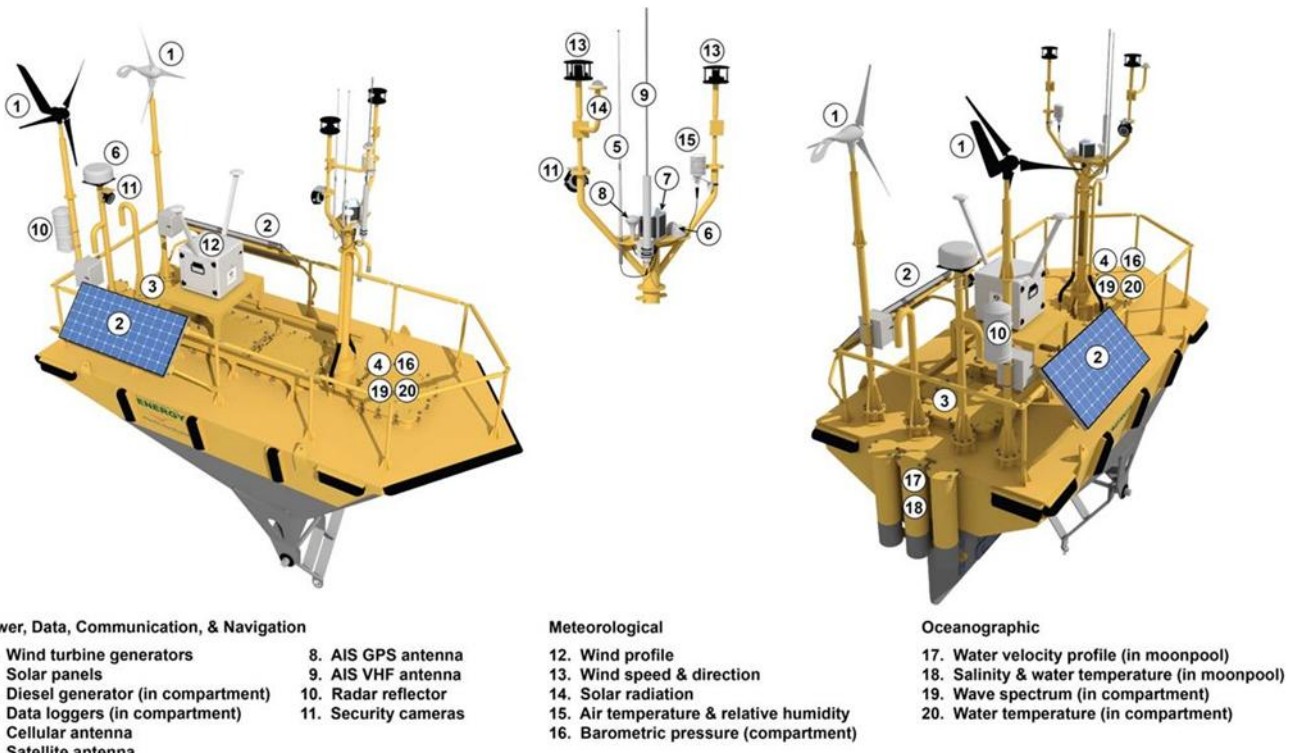

**Figure A1.** US Department of Energy Lidar Buoy schematic showing all meteorological and oceanographic observations.

**Table A1.** Sensor description.

| Reference Number | Sensor Type | Make/Model | Measurements |
|---|---|---|---|
| 12 | Wind profiling lidar | Leosphere/Windcube 866 | Vertical profile of wind speed and direction, wind dispersion, and spectral width |
| 13 | Ultrasonic anemometer (2) | Gill/WindSonic | 2D wind velocity and direction, near surface |
| 14 | Pyranometer | Licor/LI-200 | Global solar radiation |
| 15 | Temperature, relative humidity | Rotronic/MP101A | Air temperature, relative humidity |
| 16 | Barometer | RM Young/61302V | Atmospheric pressure |
| 17 | Acoustic Doppler current profiler (ADCP) | Nortek/Signature 250 | Ocean velocity and direction from sea surface to 200 m water depth |
| 18 | Conductivity temperature depth (CTD) | Seabird/SBE 37SMP-1j-2-3c | Conductivity, sea surface temperature |
| 19 | Directional wave sensor | AXYS/TRIAXYS NW II | Directional wave spectra, wave height, wave time period |
| 20 | Water temperature | AXYS /YSI | Sea surface temperature |
| N/A | Inertial measurement unit | MicroStrain/3DM GX5 45 | Yaw, pitch, roll, linear velocity, |





| (underneath the Doppler lidar) | global position, magnetometer, gyroscope |
| --- | --- |

**A2 Field deployment summary**

A 12-month measurement campaign of atmospheric and oceanographic conditions using an AXYS WindSentinel buoy was conducted off the eastern coast of Oahu, Hawaii, (Figure A2) from December 2022 through December 2023 (Table A2). Instruments on the buoy were operational throughout the field campaign, with the exception of those mentioned below. After the initial deployment the wave sensor was observed to be not operational; a service visit was completed on 16 March 2023, at 1:20 pm (HST) to repair the sensor. In addition, the lowest measurement height (40 m) of the Doppler lidar was adjusted to 50 m on 28 April 2023 due to the anomalous behaviour that was observed at 40 m range-gate. Post-deployment a thorough servicing of the lidar unit was conducted by the manufacturer, and it was determined that only the measurements from 40 m to 49 m were impacted due to an issue with the lidar's master oscillator power amplifier. Therefore, we recommend that data only above 50 m be used for any future analysis. Later in the spring of 2023, the air temperature sensor was producing occasional faulty measurements; once the weather allowed, a second service visit was completed to replace the air temperature and relative humidity sensor on 5 September 2023, at 10:00 am (HST). On 14 December 2023, 14:00 (HST), the sensors were powered down to conserve energy after receiving a low fuel warning. The wave sensor and surface wind anemometer along with the safety and navigation equipment remained powered on to track the buoy location and monitor weather conditions to find a suitable recovery window. On 15 January 2024, at 8:30 am (HST), the buoy mooring was recovered, and the buoy was towed back to shore.





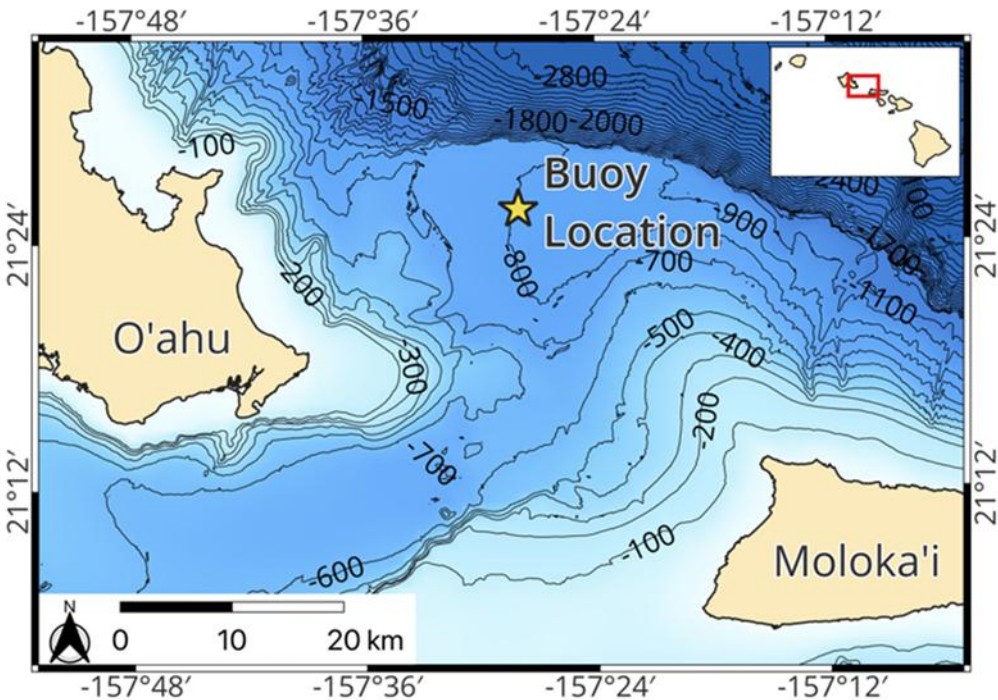

**Figure A2.** Buoy deployment location.

**Table A2.** Deployment parameters.

| Deployment location | 21.42605°N, 157.46678°W |
|---|---|
| Water depth | Approximately 815 m |
| Mooring exclusion radius | 894 m |
| Deployment start date | 1 December 2022 13:20 (HST) |
| Service visit #1 | 16 March 2023 13:20 (HST) |
| Service visit #2 | 5 September 2023 10:00 (HST) |
| Sensors powered down | 14 December 2023 14:00 (HST) |
| Deployment recovery | 15 January 2024 08:30 (HST) |

## A3 Data quality control

Throughout the field campaign, measurements collected from the buoy were transmitted via satellite communication to the DOE Wind Data Hub (http://a2e.energy.gov/project/buoy/data). Measurements from the wind profiling lidar are contained in

buoy/lidar.z07.* files; measurements from the other instruments are contained in the buoy/buoy.z07.* files; images from the onboard cameras are contained in the buoy/camera.z07.00 files. Datasets appended with *.00 are for raw data; datasets appended with *.a0 are for data processed based on automated quality control scripts. This section described the production



of the *.b0 dataset, which performs a quality control on the measurements based on post-processing and evaluation of instrument function and realistic physical phenomena. The data quality control for this field campaign was very similar to previous buoy deployments, details provided in Krishnamurthy et al., 2023.

### A3.1 Surface meteorology on the buoy

Each surface meteorological measurement (wind speed, wind direction, air pressure, air temperature, and relative humidity) was subjected to the quality control methodology of Krishnamurthy et al. (2023). At each 10-minute timestamp, the surface measurements were evaluated and determined to be of good quality, missing (due to a power outage, communication issue, or instrument availability issue), or suspect or incorrect based on typical and physical expectations (Figure A3).

During the Hawaii deployment, 5% of the surface measurements (all variables) were missing due to communication issues. At no time periods were the surface measurements deemed suspect or missing due to a buoy power outage. The portside and starboard wind speeds, portside and starboard wind directions, relative humidity, and pressure were missing due to instrument availability for less than 2% of the deployment. No instances of incorrect data were noted for the portside and starboard wind speeds, portside and starboard wind directions, and pressure, and only three instances of incorrect data (0.01%) were determined for the relative humidity, leaving 93% - 94% of good data availability for these surface variables (Figure A3).

Between 13 March 2023 and 6 September 2023, much of the air temperature data were deemed unphysical, with reported temperatures ranging between -36°C and 60°C. Due to this data quality issue, 42% of the air temperature measurements gathered during the Hawaii deployment were classified as incorrect. An additional 5% of the air temperature measurements were missing due to instrument availability, leaving 47% of the data designated with good quality and availability (Figure A3).

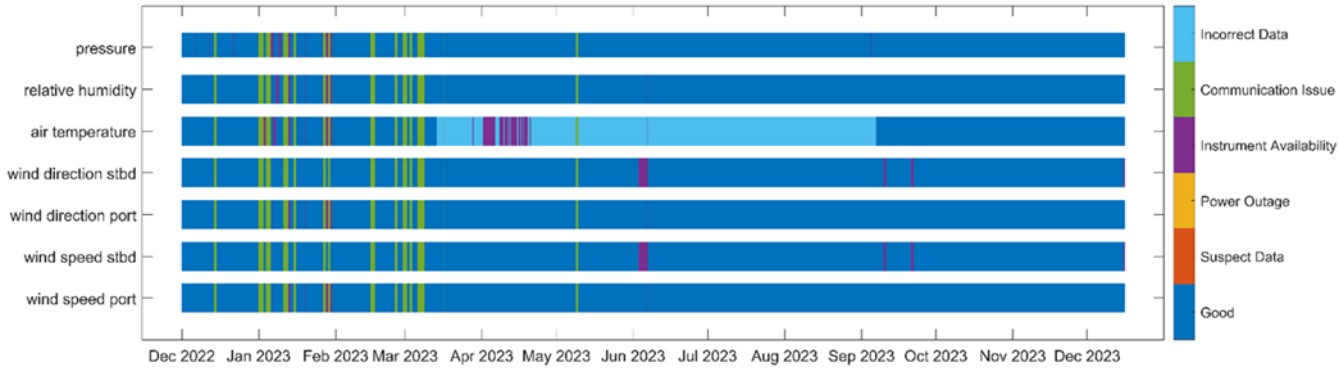

**Figure A3.** Surface meteorological data availability and quality.





In terms of capturing the temporal variability of the near surface wind speeds, the port and starboard wind speed instruments aboard the buoys were in near-perfect agreement during the Hawaii deployment, with a Pearson's correlation coefficient of 0.9993. However, a small bias (0.05 m s$^{-1}$) exists between the two instruments during the period of overlapping data availability, with the port wind speed measurements being slightly faster. Similarly, the portside wind directions are on average 3° greater than the starboard wind directions.

**A3.2 Buoy Doppler lidar**

Height-resolved measurements of wind speed, wind direction, and turbulent intensity were provided by a Windcube v2 Doppler lidar, as shown in Figure A1. This buoy and lidar had previously been deployed off the California coast near Humboldt Bay. For the Hawaii deployment, the Windcube was initially configured in the same manner, with 12 measurement heights ranging every 20 m from 40 to 240 m (Krishnamurthy et al., 2023). The lowest measurement height was adjusted up to 49 m on 28 April 2023 due to suspected bias in the results at 40 m AGL, as mentioned previously. The heights of the other 11 measurement heights remained unchanged throughout the deployment.

The raw 1-Hz wind profiles measured by the Windcube were corrected for platform motion. For the current deployment, we use the same motion-correction procedure that was applied during a previous deployment near Humboldt Bay, California (Krishnamurthy et al., 2023). This procedure uses input from an externally mounted backup inertial measurement unit (IMU) to essentially bypass the lidar's internal IMU which had malfunctioned. Further details about the motion-correction process are provided by Krishnamurthy et al. (2023).

Motion-corrected 1-Hz data are used to compute ten-minute averages of wind speed, wind speed variance, wind direction, wind direction variance, as well as velocity variances and covariances. Prior to averaging, the 1-Hz data were filtered to remove poor quality samples with carrier-to-noise-ratios (CNR) below –23 dB. Velocity variances were computed by first linearly detrending the 1-Hz data (Krishnamurthy et al., 2023). The data availability was also computed as the percentage of 1-second samples above the CNR threshold (-23 dB). We note, however, that there was a small amount of deadtime associated with the IMU data. Raw 10-Hz IMU were stored in half-hourly files, which were not temporally contiguous. There was about a 30-second gap at the beginning and end of each raw IMU file. This resulted in a 1-minute gap every 30-minutes. As a result, the data availability is ultimately limited by these short gaps in the raw IMU data.

Figure A4 shows the profile of lidar data availability (DA) during the Hawaii deployment. The data availability represents the percentage of time that valid 1-Hz measurements are available within each 10-minute averaging period. It is important to note that the DA is computed over the time for which valid data exist. As a result, the DA for the two lowest measurement heights were computed over shorter time periods than the other heights. The data availability generally degrades with altitude, particularly above about 150 m AGL. The height-averaged DAs was 98%. As noted above, the DA is ultimately limited by the dead time in the raw 10Hz IMU data.

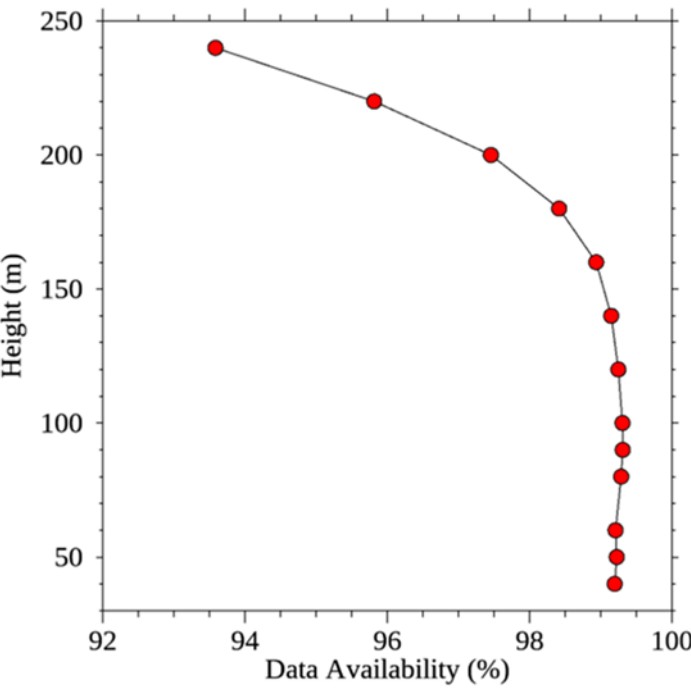

**Figure A4.** Data availability for the lidar winds during the Hawaii deployment. The data availability for the lowest measurement height
(40 m) was computed over the period from 1 December 2022 to 28 April 2023, post which the measurement at this height was not
recorded anymore due to an issue observed at this height. The data availability at 50 m was computed over the period from 29 April to 14
December 2023. The data availability at 60 m and above was computed over the period from 1 December 2022 to 14 December 2023.

The wind speed profiles shown in Figure A5a exhibit very little shear above the 50 m. At 100 m AGL the median wind
speeds range from about 8.5 to 9.5 m s$^{-1}$, depending on the averaging period. This is significantly smaller than the wind
resources observed by this same lidar off the California coast (Krishnamurthy et al., 2023). Also shown in Figure 5a are
profiles of the median wind speed during periods with positive and negative air-sea temperature differences. Here the air-sea
temperature is used to indicate stable versus unstable conditions. The air-sea temperature difference was obtained from
difference between air temperature sensor at ~3 m, and the CTD water temperature sensor at a water depth of ~1 m. We
found that the air-sea temperature differences to be negative (unstable) 64% of the time during the deployment period.

Shown in Figure A5c are profiles of the difference between the motion-corrected and uncorrected wind speed and wind
direction. The wind speed differences are quite small, with the motion-corrected wind speeds only about 2 mm s$^{-1}$ faster
than the uncorrected winds. By contrast, motion-correction had a significant effect on the wind direction, as one would
expect. The lidar wind direction profiles shown in Figure A5 indicate a strong preference for easterly flow, and no
significant rotation with height within the first 250 m above the sea surface.



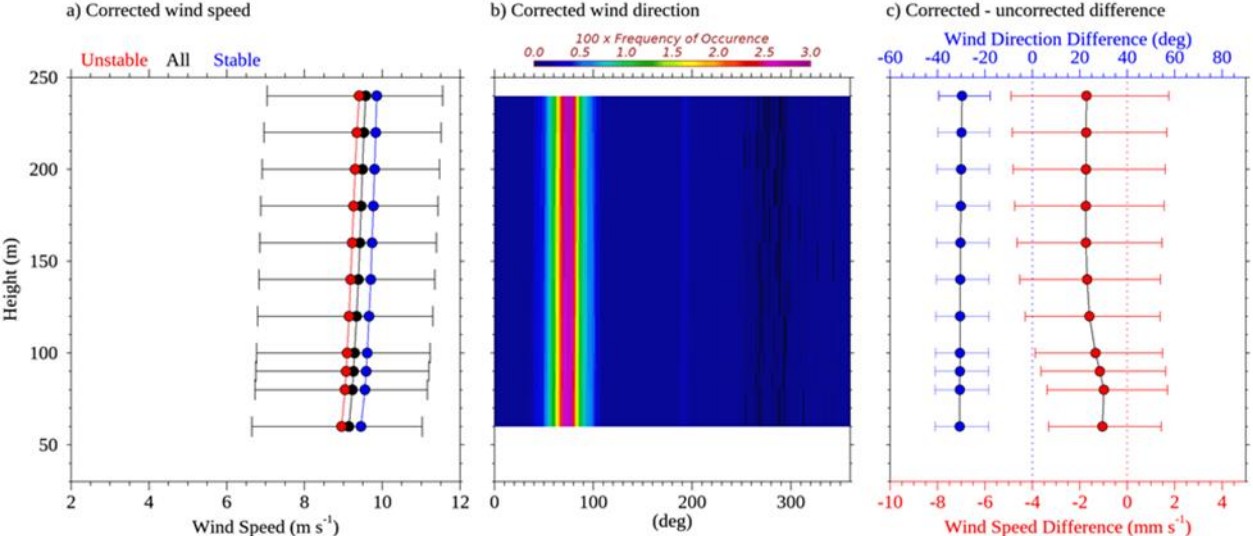

**Figure A5.** Results from the Windcube v2 averaged from 29 April 2023 to 14 December 2023 showing (a) the median corrected wind speed profile (black), b) the corrected wind direction distribution profile, and (c) the difference between the motion-corrected and uncorrected wind speed and wind direction. Also shown in (a) are the median wind speed profiles for periods with positive (blue) and negative (red) air-sea temperature differences. Error bars show the 25$^{th}$ to 75$^{th}$ percentile range.

### A3.3 Buoy oceanographic measurements

Surface gravity wave (i.e., wave) data were measured with the TRIAXYS sensor at a 20-minute sampling interval, the typical standard for wave measurements in the United States (NDBC, 1996). Wave measurements began on 16 March 2023, following the first service visit and ended with the deployment recovery on 15 January 2024. Initial quality control checks (i.e., included in *.a0 files on the Wind Data Hub) follow AXYS specifications and remove erroneous data by comparing individual measurements to temporally adjacent ones and removing all data when significant wave height reads larger than 40 m (AXYS, 2012). All data measured by the TRIAXYS wave sensor are given in Table A3. The mean wave period is calculated from the zeroth and first spectral moments of the wave spectra, differing from average wave period, $T_{avg}$, which is a mean of the time series. Peak wave period and peak wave direction are also calculated from the wave spectrum and not directly measured. AXYS post-processing software (version 5.01) is used to calculate the wave spectrum, which applies the maximum entropy method (Nwogu, 1989).

A nearby National Data Buoy Center (NDBC) buoy (#51202: Mokapu Point) was used to cross-check the lidar buoy wave measurements. Station 51202 is 21 km from and ~700 m shallower (depth = 86 m) than the Oahu lidar buoy location. The shallower depth allows some wave frequencies to transition from deep-water (i.e., $D > L/2$, where $D$ is depth and $L$ is wavelength) to transitional waves (i.e., $L/20 < D < L/2$). A comparison of significant wave heights between locations outlines largely similar wave climates, though the lidar buoy recorded wave heights slightly larger offshore than at 51202 (Figure A6). Multiple factors could create this variability: deep water – transitional wave transformation, as well as variability in





local bathymetry and wind fields; highlighting the importance of local measurements for wave applications. The minimum
620  significant wave height measured at 51202 for the deployment period was 0.6 m, so significant waves heights less than 0.25
m at the lidar buoy were flagged as bad. If significant wave heights were marked bad for exceeding 40 m or failing to
surpass 0.25 m, the remaining variables were also marked bad, as they are derived from the same sensor.

An additional quality check was applied to identify possible rogue waves. A ratio of maximum wave height to significant
wave height ($H_{max}/H_{sig}$) greater than 2 is typically considered indicative of a rogue wave (e.g., Müller et al., 2005; Nikolkina
625  & Didenkulova, 2011). Data points where rogue waves are possible would be marked as questionable, but we found no data
in this deployment to exceed that criterion. Table A3 outlines the total number of good, questionable, and bad data for each
measurement during the 16 March 2023, to 15 January 2024, wave sampling period. Other gaps in the wave and ocean
observations were potentially due to a sensor failure and were not post-processed.

630  **Table A3:** TRIAXYS wave sensor measurements and number data marked good, questionable, or bad. Variables short names are provided
to match those in *.b0 files, if applicable.

| Wave measurement | Good | Questionable | Bad |
|---|---|---|---|
| Number zero crossings, $ZCN$ | 21,823 | 0 | 16 |
| Average wave height, $H_{avg}$ | 21,817 | 0 | 22 |
| Average wave period, $T_{avg}$ | 21,817 | 0 | 22 |
| Maximum wave height, $H_{max}$ | 21,815 | 0 | 24 |
| $10^{th}$ percentile wave height, $H_{10}$ | 21,817 | 0 | 22 |
| $10^{th}$ percentile wave period, $T_{10}$ | 21,817 | 0 | 22 |
| Significant wave height, $H_{sig}$ | 21,823 | 0 | 16 |
| Significant wave period, $T_{sig}$ | 21,817 | 0 | 22 |
| Mean wave direction | 21,817 | 0 | 22 |
| Mean wave spread | 21,817 | 0 | 22 |
| Mean wave period | 21,817 | 0 | 22 |
| Peak wave direction | 0 | 0 | 21,839 |
| Peak wave period | 21,817 | 0 | 22 |



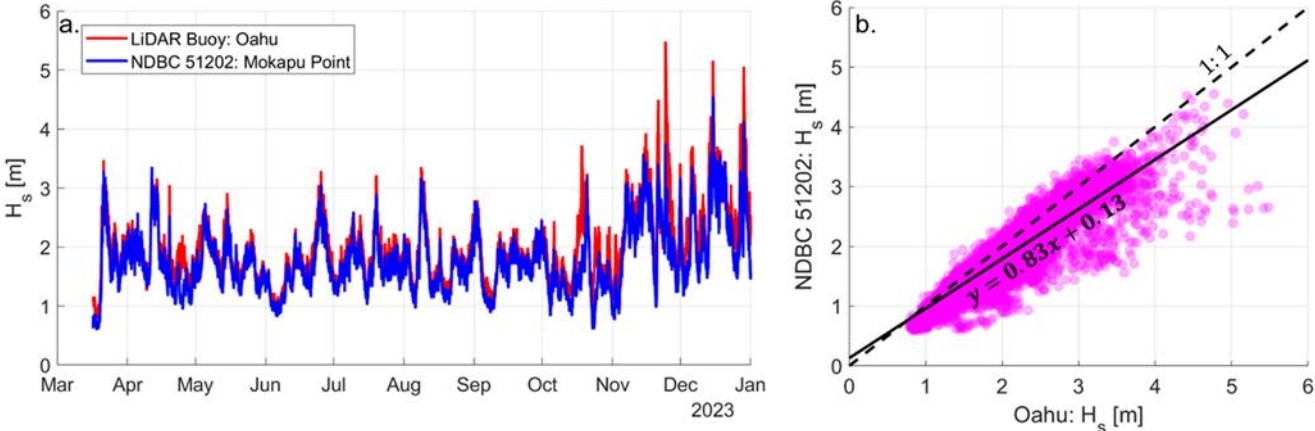

**Figure A6:** (a) Significant wave heights measured at the lidar buoy (red) and NDBC station 51202 (blue) from 16 March 2023 to 31 December 2023. (b) Scatter comparison of significant wave heights from each station at each time step. A linear regression best fit is also shown.

Ocean currents were measured with the Nortek Signature 250 acoustic doppler current profiler (ADCP) using 10-minute ensemble averaging. The ADCP recorded measurements from the deployment start on 1 December 2022, through sensor power-down on 14 December 2023. Specifically, velocity magnitude and direction were measured at 50 depth bins, vertically spaced 4 m apart. Current magnitudes and direction were marked bad if: data was missing, spikes in current magnitude occurred that were spatially and temporally uncorrelated over a 10 min. duration, and/or a measurement was isolated in time (i.e., at least two successful measurements before and after did not occur). Current data were marked questionable if vertical shear in current speed was larger than 0.2 m s$^{-1}$ and/or a vertical measurement was isolated in space (i.e., at least two successful measurements above and below did not occur). The ADCP sampling range typically reached 100 to 120 m in depth, with the deepest measurements at 141 m. Measurements below the sampling range are considered missing data and therefore marked as bad. In total, 999,158 current measurements were marked good, 80,261 were marked questionable, and 1,511,731 were marked bad.

Conductivity was measured with the Sea-Bird CTD and sea surface temperature (SST) from both the Sea-Bird CTD and YSI thermistor every 10 minutes over the same duration as the ADCP (1 December 2022 to 14 December 2023). CTD data were marked bad for either instrument if a spike in either SST or conductivity occurred which was temporally uncorrelated to surrounding data points. No "questionable" checks were applied to the CTD measurements. In total, the Sea-Bird CTD recorded 22,018 good and 29,805 bad conductivity and SST measurements. The YSI thermistor recorded 44,305 good and 7,518 bad SST measurements.



### A3.4 Buoy pyranometer

In Krishnamurthy et al., (2023), we discussed the acquiring of coastal cloud properties obtained from broadband global solar radiation (GSR) data using the pyranometers (PYRs) deployed on the buoy. The difference between the measured surface irradiance and its estimated clear-sky counterpart (Figure A7a) serves as the foundation for determining cloud optical thickness (COT, Figure A7b) and generating the corresponding cloud mask (Figure A7c). The COT characterizes the total reduction of downwelling solar radiation due to cloud droplets and/or ice crystals, and a dense cloud with a significant COT has a visually "dark" appearance to an observer on the surface. Thus, a large difference between the measured and clear-sky irradiances typically indicates the presence of a dense cloud above a pyranometer. Conversely, small differences suggest the chance of optically thin clouds. It is noteworthy that small differences may also be attributed to dense aerosol plumes originated by various sources, such as the somewhat unique Hawaiian feature of vog, or also by fogs. Generally, optical thickness values for both aerosol plumes and fogs remain below 4. Consequently, the calculated COT values exceeding 4 provide greater confidence of cloud presence above the pyranometer, while lower values suggest otherwise (Figure A7b). In other words, the COT can be viewed as a form of quality assurance (QA) for assessing cloud properties, such as a binary cloud mask (Figure A7c). We also perform supplementary QA checks to identify days where the PYR-measured GSR data exhibit incomplete daytime coverage, featuring extended gaps lasting up to several hours. Cloud properties are not provided for these identified days.

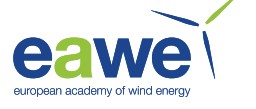
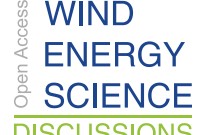

**Figure A7:** (a) The GSR measured (OBS) for a given day (18 July 2023) and location (Hawaii) and its estimated (or model) clear-sky (MOD) counterpart; (b) calculated COT and (c) estimated cloud mask.


**Data availability statement**

The lidar buoy data utilized in this study are freely and publicly available from the U.S. Department of Energy. The Hawaii lidar wind dataset is available at https://doi.org/10.21947/2335446 and the accompanying near surface observations at https://doi.org/10.21947/2328673. ERA5 is provided via the Copernicus Climate Data Store at 680 https://cds.climate.copernicus.eu/. UH-WRF was developed by the University of Hawaii and shared with Pacific Northwest National Laboratory through a research partnership. NOW-23 is hosted by the National Renewable Energy Laboratory's





Wind Resource Database at https://wrdb.nrel.gov/data-viewer. Global Wind Atlas is available at https://globalwindatlas.info/en/.

## Author contributions

LS is responsible for conceptualization, data curation, formal analysis, and writing. RK provided conceptualization, supervision, and writing. RN, PS, and EK contributed data quality control and writing. TN, YC, FS, and NB developed some of the simulations used in this work and reviewed the manuscript. WG and YL supported with data access, processing, and manuscript review. MP and MS were critical to the success of the observational campaign that provided the measurements used in this work.

## Competing interests

The authors have no competing interests to declare.

## Acknowledgements

This work was authored by the Pacific Northwest National Laboratory, operated for the U.S. Department of Energy by Battelle (contract no. DE-AC05-76RL01830). This work was authored in part by the National Renewable Energy Laboratory
for the U.S. Department of Energy (DOE) under Contract No. DE-AC36-08GO28308. Funding provided by U.S. Department of Energy Office of Energy Efficiency and Renewable Energy Wind Energy Technologies Office. The views expressed in the article do not necessarily represent the views of the DOE or the U.S. Government. The U.S. Government retains and the publisher, by accepting the article for publication, acknowledges that the U.S. Government retains a nonexclusive, paid-up, irrevocable, worldwide license to publish or reproduce the published form of this work, or allow
others to do so, for U.S. Government purposes.

The authors would like to thank the U.S. Department of Energy Wind Energy Technologies Office for funding this research. PNNL would also like to thank the Wind Data Hub Team, especially Kenneth Burk, Max Levin, Chitra Sivaraman, Matthew McDuff, and Sherman Beus. The team would also like to acknowledge the buoy contractor, AXYS technologies, for their support maintaining the buoys and for data verification during the deployment and Raj Rai, Amy Brice, and James
Marquis for reviewing the initial draft article.

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
