# Peer review of "Performance of reanalysis and mesoscale models for wind resource assessment off the coast of Hawaii"

_Wind Energy Science, 2025_

## Referee Comment (RC1)

This is a well written and important paper that evaluates the statistics of near-surface wind at an offshore site off Hawaii relative to lidar buoy measurement of the vertical wind profile. Given the sparsity of offshore wind observations, this is an important study, even though it is constrained to just a single observation site. Moreover, the location sits in the unique, persistent tradewind environment, so is remarkably undisturbed by topographic or other continental features. I do however think that there are a number of aspects of the meteorology of the region that have been overlooked, and addressing them would add to the usefulness of this paper. These aspects are described below, followed by a series of minor comments.

**1. Interannual variability**
Given the position of Hawaii in the central Pacific Ocean and the fact that most of the study is based on a single year of data, it seems remiss to ignore the interannual variability. 2023 was an El Nino year, and was reported to have high rainfall across the pacific (eg https://www.drought.gov/sites/default/files/2023-06/Pacific%20Spring%202023.pdf). Moreover, there is a strong influence of ENSO on the strength and extent of the tradewinds. This means that when comparing the 2023 dataset to the 2000-2019 and 2008-2017 NOW-23 and GWA3 datasets, there will be a particular bias resulting from the number cases of each ENSO phase. I understand that the dataset of 2023 observations and the high-resolution WRF simulation are limited, but I think some analysis of the impact of the interannual variability on this site would be helpful in establishing whether 2023 was a 'unusual' year. This could be done using the ERA5 dataset over a longer time period. A further suggestion would be to try restricting the 2000-2019 and 2008-2017 datasets to just the El Nino years, to give a more realistic base for comparing the 2023 data to.

**2. Extreme wind influences on the annual cycle**
In addition to the systematic interannual variability signal, there are likely some extreme events that contributed to the variability of the 2023 dataset. For example, there were two Kona low events reported in February 2023, which may go a long way to explaining the spike in wind speeds seen in figure 5. Eg see https://www.weather.gov/hfo/KonaLowFeb2023 and the associated MSLP map for 17 Feb 2023 (below).. This is actually quite a fascinating situation, and assessing whether the reanalysis products were able to accurately capture the strong wind associated with these events is an important diagnostic in itself. It doesn't really make sense to talk about the "steep observed monthly wind increase of 5.66 m/s between January and February 2023" (line 268)  as a statistical feature when there was a distinct and unusual weather event causing this difference. Similarly, it doesn't make sense to compare the 2023 monthly averages with the GWA3 and NOW-23 datasets over different time periods, when what we're seeing is less a systematic difference and more the influence of a single weather event. Some relevant points that could be address here are: How unusual were the 2023 February Kona wind events, relative to the time periods in the other datasets? How do the datasets compare if you restrict the analysis to days that just had a 'classic' tradewind regime?

[Figure]

00Z 17 Feb 2023  1000 hPa                                            University of Wyoming
https://weather.uwyo.edu/cgi-bin/uamap?REGION=npac&OUTPUT=gif&TYPE=obs&TYPE=an&LEVEL=1000&date=2023-02-17&hour=0

**3. Local aspects**

A great strength of this study is the fact that the wind is almost undisturbed tradewind flow. However, there may be some local influences relating to the effects of the islands. This could be especially relevant to the diurnal cycle in figure 13, and potentially particularly in the rare cases when the wind is blowing from the ESE or SSW. The lidar observations have a larger-amplitude diurnal cycle than either of the reanalysis datasets studied, but is this due to the different prevalence of wind directions in the different datasets? I suggest splitting the diurnal cycle up into different wind direction bins, although I know it will be difficult to get statistical significance for the rarer directions. A relevant reference for interaction of local phenomena with the tradewind environment is Dao et al. (2025):

Dao, T.L., Vincent, C.L., Huang, Y., Peatman, S.C., Soderholm, J.S., Birch, C.E., et al. (2025) Joint modulation of coastal rainfall in Northeast Australia by local and large-scale forcings. *Quarterly Journal of the Royal Meteorological Society*, e70027. Available from: **https://doi.org/10.1002/qj.70027**

**4. Treatment of the short dataset**

As already mentioned before, the treatment of the short dataset is not quite convincing. The authors should use the ERA5 dataset to establish the links between the other datasets that all cover different periods: Eg. a plot of the monthly wind averages from ERA5 for the periods 1950-present, 2000-2019, 2008-2018 and 2023 will help tie everything together, and clarify the representivity of the different time periods. This may indicate some further filtering as mentioned above, such as restricting the analysis to El Nino years.

**Specific Comments**

Lines 57-80. Could this information be presented in a more integrated way? Listing them one by one is difficult to read and doesn't leave any overall impression of the issues. Perhaps group by dataset tyoe: "Several studies have evaluated ERA5 at offshore sites, and noted a low wind speed bias (ref ref). etc". I guess the key message is that there has only been a small number of studies so far, the results are inconsistent across the studies, which is what makes this current study important.

Line 92-92 – Define YSU and MYNN acronyms

Line 104 – is **metocean** actually a word?

Line 136 – "off of Oahu" -> "off Oahu"

Figure 2 – Where does the 4 metre observation come from?

Line 177 – 'EMCWF' -> 'ECMWF'

Line 191 – Presumably the 1.5 km simulation didn't use a cumulus parametrisation?

Table 1: The GWA3 dataset should mention the microscale downscaling, not just the mesoscale modelling part

Equations 1-3. Are these standard equations really necessary? Perhaps check with the editors.

Figure 3: It would be better to make the time period for these plots the same (1 Jan 2023 – 15 December 2023).

Figure 4 and accompanying text: Define that you're talking about the median bias, not the mean bias.

Line 284 – 'gently' -> 'smoothly'

Figure 5 – perhaps a bar chart would be more appropriate, since this is not relaly continuous data.

Line 293. Clarify what these correlations are. Are they between the 24 values of the average diurnal cycle for each dataset?

Figure 7 – I think it would be clearer to just show the observed wind rose.

Figure 8 – It might be better to calculate the Richardson number or other stability metric, rather than just the air-sea temperature difference.

Section 4.1. I'm really not sure that it's reasonable to talk about bias here, given the different time periods. See general comments. Use the same time period for ERA5, NOW23 and GWA3, and then also do it for just 2023 for ERA5.

Figure 11 – What do the box plots show? Is it all values in the 2000-2019 period, or the distribution of the 20 annual averages during that time?

Figure 13 - Why were the ERA5 and WRF datasets not included in the diurnal cycle study?

---

## Referee Comment (RC2)

This study evaluates the performance of reanalysis, regional mesoscale simulations, and purpose-built wind datasets in a promising yet sparsely observed offshore environment near Hawaii. Using a year-long lidar buoy deployment off the eastern coast of Oahu, it characterizes the local wind resource and validates four datasets (ERA5, UH-WRF, NOW-23, GWA3) at different spatial resolutions. The region's persistent trade winds make it favorable for wind power, but limited hub-height observations underscore the need for accurate data. By comparing multiple products in this uniquely characterized setting, the work is relevant to the sector and appears suitable for publication pending the revisions below.

**Major concerns**

**1. Attribution to PBL choice**

The manuscript notes differences in PBL schemes across datasets but does not isolate the PBL contribution with, for instance, a sensitivity test. The study even presents several diagnostics informative of boundary-layer processes (e.g., stability-stratified errors, diurnal cycle, vertical shear), but the connection from these results to PBL-scheme attribution is not made explicit. If this is the case, please state explicitly how each diagnostic reflects expected PBL behavior, or soften the claim and note that other factors are in play.

**2. Shifting focus from validation to campaign report**

The buoy campaign is essential for the validation, but exhaustive operational/installation details distract from the paper's focus. Keep only what is needed for readers to understand, reproduce, and trust the results (site, period, completeness, QC, uncertainties, key post processing). Technical but relevant info can go in the Appendix, but non-essential logistics/engineering details should be moved to a separate deployment report.

**Minor revisions**

L89: Several times in the text, the word "trend(s)" could be replaced with "result(s)", "pattern(s)" or "behavior".

L92: Define MYNN and YSU on first use.

L93: Remove "using MYNN".

L97: "Nunalee and Basu (2014) noted …" — consider removing if it does not add to the argument in that paragraph.

L101: Is there a reference for the observational campaign (e.g., Krishnamurthy et al., 2023) and/or others; point to Appendix A for site-specific details.

L103: Replace "hitherto have not been explored in an observational manner" with "had not previously been observed."

L114: "…ERA5 will exhibit a low wind speed bias …" this entire sentence is ambiguous.

Section 2: Consider merging the opening of Section 2 with the the first paragraph of Section 2.1 to present a concise summary of the observational data (source, location, period, sampling/averaging, heights, instrument/method). The rest could be moved to a new subsection (e.g., Local Wind Characterization). You could briefly summarize the key points and refer to a single comprehensive figure that combines the already shown plots 2a and 2b, and adds (c) monthly and seasonal means and (d) the diurnal cycle, rather than exhaustively listing numbers. Any additional technical details should go in the Appendix.

Section 2.2: Clarify whether the ERA5 time series was horizontally interpolated or taken from the nearest sea or land/sea grid cell. The proximity to the coastline could influence results.

L179: Provide a reference for UH-WRF if available.

L205: Clarify that GWA3 wind data are provided as annual, monthly, and diurnal climatologies, not continuous hourly time series.

Table 1: Provide actual vertical levels used rather than "lowest N model heights." For ERA5, "PBL scheme plus the effect of data assimilation"?

L238–239: It appears the manuscript uses "average/median bias" when referring to mean/median errors. It sounds a bit redundant as bias is already the mean error. Unless you compute multiple bias values (e.g., per height or per wind speed range) and then take their mean/median. Remove the word "average" in both lines if redundant and check the entire manuscript.

L241–243: "Figure 3a".

Figure 3: There is no plot 3d.

Fig. 5: State normalized by what and its relevance in the text.

Section 3.2: Add one sentence explaining why testing different atmospheric stability conditions is relevant. Also, state explicitly that you use air-sea temperature differential as a proxy for stability and note the limitations.

L358: Include results/discussion on occurrences of LLJs, or remove earlier references if not analyzed.

L368: Reference figure 11a.

L383: Specify inconsistencies relative to what.

L433: Use "corroborate previous reports of" instead of "add another geographic data point to the trend of".

L437–438: The winds may be consistent, but was 2023 a regular year? Maybe it would be helpful to include this information. Also, what is the distance from the buoy to the nearest coastline? If the ERA5 grid cell includes or lies close to the coast, this proximity could influence the results.

L453–455: Justify attributing UH-WRF's smaller bias primarily to the PBL scheme (YSU) rather than to other differences.

L487: There is no Shaw et al., 2020 in the references. Do you mean Gorton and Shaw, 2020?

Appendix A1: Keep only information that adds to the manuscript and is not already documented elsewhere: location, measurement method, campaign duration, data completeness, uncertainties, QC, and post-processing.

Appendix A3.3 and A3.4: Consider removing if not essential to understanding the manuscript.

---

## Author Comment (AC1)

This is a well written and important paper that evaluates the statistics of near-surface wind at an offshore site off Hawaii relative to lidar buoy measurement of the vertical wind profile. Given the sparsity of offshore wind observations, this is an important study, even though it is constrained to just a single observation site. Moreover, the location sits in the unique, persistent tradewind environment, so is remarkably undisturbed by topographic or other continental features. I do however think that there are a number of aspects of the meteorology of the region that have been overlooked, and addressing them would add to the usefulness of this paper. These aspects are described below, followed by a series of minor comments.

Thank you very much for the time and effort you put in to read our paper and suggest improvements. We are grateful for all the ideas you contributed and have addressed them throughout the manuscript as follows. Again, thank you!

1. Interannual variability

Given the position of Hawaii in the central Pacific Ocean and the fact that most of the study is based on a single year of data, it seems remiss to ignore the interannual variability. 2023 was an El Nino year, and was reported to have high rainfall across the pacific (eg https://www.drought.gov/sites/default/files/2023-06/Pacific%20Spring%202023.pdf). Moreover, there is a strong influence of ENSO on the strength and extent of the tradewinds. This means that when comparing the 2023 dataset to the 2000-2019 and 2008-2017 NOW-23 and GWA3 datasets, there will be a particular bias resulting from the number cases of each ENSO phase. I understand that the dataset of 2023 observations and the high-resolution WRF simulation are limited, but I think some analysis of the impact of the interannual variability on this site would be helpful in establishing whether 2023 was a 'unusual' year. This could be done using the ERA5 dataset over a longer time period. A further suggestion would be to try restricting the 2000-2019 and 2008-2017 datasets to just the El Nino years, to give a more realistic base for comparing the 2023 data to.

We really appreciate this great idea and feel that adding the interannual context at your recommendation has improved the analysis significantly. Per your suggestion, we analyzed the most recent 40 years of ERA5 wind, precipitation, and temperature data, along with the Oceanic Niño Index, to set the stage for what kind of year 2023 was at the location of the lidar buoy deployment.

We now begin Section 3 as follows. Lines 180-189: "Prior to comparing the O'ahu observations, which predominantly occur in the year 2023, with atmospheric datasets, it is imperative to provide context on what kind of meteorological year 2023 is relative to the long-term interannual wind speed variability noted at the deployment location. Utilising annual averages of the Oceanic Niño Index (ONI), we find that 2023 is categorized as an El Niño year based on the Climate Prediction Center's threshold of +/- 0.5°C (Figure 3a) (NOAA, 2025). In Figure 3b, we explore the annual average ERA5 100 m wind speeds (CDS, 2025b) between 1985 and 2024 and determine that 2022 and 2023 are tied for having the lowest annual average wind speeds over the 40-year period (annual average wind speeds normalized by the 40-year mean for both years = 0.88) while being opposingly classified as La Niña and El Niño years, respectively (Figure 3a). According to ERA5, precipitation and 2 m temperature are above average at the buoy location for the year 2023 (Figure 3c, d). The above average precipitation, temperature, and weakening of the trade winds are consistent with expected El Niño characteristics (Lu et al., 2020)."

[Figure]

**Figure 3.** (a) Annual average ONI, (b) normalized annual average 100 m wind speed from ERA5, (c) normalized annual total precipitation from ERA5, and (d) normalized annual average 2 m temperature from ERA5 over the 40-year period between 1985 and 2024 coloured by the annual average ONI. The annual average wind speeds are normalized by the 40-year average wind speed at the O'ahu buoy deployment location.

This analysis, which we appreciate you suggesting and encouraging, guided our decision that given the atypical wind resource year that 2023 was, we could not appropriately analyze the non-concurrent datasets NOW-23 and GWA3. Leading up to such a decision, we took your helpful suggestion to assess the ERA5 during 2023 relative to the coverage periods of NOW-23 (2000-2019) and GWA3 (2008-2017).

As can be seen in Figure 3b, the low annual average wind speeds during 2023 are not replicated by any other year that falls within the NOW-23 or GWA3 coverage period, which means we cannot make any statements concerning the performance of these datasets using the 2023 lidar buoy observations. We make mention of this on Lines 190-194: "The following analyses focus on the performance of two simulation datasets that have concurrent temporal coverage with the 2023 observations: ERA5 and UH-WRF. Given the context of 2023 being a record low wind resource year east of O'ahu according to the ERA5 record, it is important to note the need for long-term, continuously updating datasets like reanalyses. Purpose-built wind datasets, like NOW-23 (2000-2019) and Global Wind Atlas (2008-2017), provide numerous years for wind resource assessment but would not represent the characteristics of an atypical year like 2023."

Lu, B-Y., Chu, P-S., Kim, S-H., and Karamperidou, C.: Hawaiian Regional Climate Variability during Two Types of El Niño, Journal of Climate, 33, 22, 9929-9943, https://doi.org/10.1175/JCLI-D-19-0985.1, 2020.

NOAA (National Oceanic and Atmospheric Administration): Climate Variability: Oceanic Niño Index, https://www.climate.gov/news-features/understanding-climate/climate-variability-oceanic-nino-index, last updated 25 June 2025.

**2. Extreme wind influences on the annual cycle**

In addition to the systematic interannual variability signal, there are likely some extreme events that contributed to the variability of the 2023 dataset. For example, there were two Kona low events reported in February 2023, which may go a long way to explaining the spike in wind speeds seen in figure 5. Eg see https://www.weather.gov/hfo/KonaLowFeb2023 and the associated MSLP map for 17 Feb 2023 (below). This is actually quite a fascinating situation, and assessing whether the reanalysis products were able to accurately capture the strong wind associated with these events is an important diagnostic in itself. It doesn't really make sense to talk about the "steep observed monthly wind increase of 5.66 m/s between January and February 2023" (line 268) as a statistical feature when there was a distinct and unusual weather event causing this difference. Similarly, it doesn't make sense to compare the 2023 monthly averages with the GWA3 and NOW-23 datasets over different time periods, when what we're seeing is less a systematic difference and more the influence of a single weather event. Some relevant points that could be address here are: How unusual were the 2023 February Kona wind events, relative to the time periods in the other datasets? How do the datasets compare if you restrict the analysis to days that just had a 'classic' tradewind regime? https://weather.uwyo.edu/cgi-bin/uamap?REGION=npac&OUTPUT=gif&TYPE=obs&TYPE=an&LEVEL=1000&date=2023-02-17&hour=0

We thank you for this suggestion to investigate extreme wind influences and have addressed your helpful recommendation as follows:

First, we added a new section that covers three wind events (Section 3.7 Extreme weather events), including the Kona lows that you pointed out. We found that the wind speeds at the lidar buoy location were weaker during the Kona low periods, with no extreme wind events, and stronger during the typical trade wind occurrences during the month of February 2023. Lines 406-423:

"**3.7.1 February 2023 Kona lows**

Two consecutive Kona lows (Morrison and Businger, 2001) developed near the Hawaiian Islands in mid-February 2023, resulting in heavy rainfall impacts on several islands (NOAA, 2023a). The buoy observations during this period confirm the disruption of the easterly trade winds, with wind directions becoming more variable and wind speeds weakening (Longman et al., 2021). No extreme wind events are recorded by the buoy during the Kona low events; rather, the wind speeds are elevated during the trade wind dominant periods before and after the events (Figure 15a).

   While both simulation datasets capture the decrease in wind speed (Figure 15a) and shift in wind direction from the dominant easterly pattern (Figure 15b), they struggle with representing the temporal variations in the wind speeds during the Kona low events. ERA5 and UH-WRF produce notably lower correlations during the low events between 15-19 February (0.63 and 0.46) relative to the entire deployment (0.89 and 0.85). The ERA5 wind speed biases at 140 m are smaller during the Kona low events relative to the entire deployment (-1.34 m s$^{-1}$ versus -1.54 m s$^{-1}$), while UH-WRF overestimates the observed wind speeds during the events at a greater magnitude than the model underestimates the observed wind speeds during the entire deployment (0.46 m s$^{-1}$ versus -0.25 m s$^{-1}$). Following the Kona low on 19 February, ERA5 simulates lower wind speeds for an extended period (19-25 February) compared to the observations. This could be attributed to several factors, including challenges in data assimilation after the Kona event, residual atmospheric instabilities delaying boundary layer recovery, misinterpretation of sea surface temperatures and moisture fluxes, or limitations in model physics and parameterizations. Further analysis will be addressed in upcoming work."

[Figure]

**Figure 15.** Observed and modelled (a) wind speeds and (b) wind directions and (c) observed 4 m air pressure at the lidar buoy location during February 2023.

Next, we took your recommendation to examine the long-term ERA5 dataset in terms of the seasonal cycle. We found that the unique pattern of the monthly average wind speeds in 2023 relative to 40 years of ERA5 wind speeds mirrors the pattern of the frequency of occurrence of the trade winds in 2023 relative to the same 40 years.

Lines 231-249: "The seasonal wind cycle during the lidar buoy deployment is characterized by immense variability in the fall and winter and more static conditions in the spring and summer (Figure 6). The observed monthly 140 m wind speed standard deviation is 2.31 m s$^{-1}$ for fall and winter months (September – February) and 0.79 m s$^{-1}$ for spring and summer months (March – August). Especially notable is the steep observed monthly 140 m wind speed increase of 5.66 m s$^{-1}$ between January and February 2023 (Figure 6a); however, it is important to explore the 2023 seasonal patterns in the context of the long-term patterns. Looking at the monthly wind speeds normalized by the annual average wind speed from the 2023 deployment observations and the simulation datasets, which show similar patterns, in the context of 40 years (1985 to 2024) of normalized monthly 100 m wind speeds from ERA5 (CDS, 2025b), the 2023 seasonal wind speed cycle is atypical relative to the long-term (Figure 6b). While weather events can occur along the eastern shores of O'ahu that alter the typical conditions (Morrison and Businger, 2001) and potentially the temporal wind speed patterns (see Section 3.7), long-term ERA5-based analysis indicates that

the 2023 seasonal wind speed cycle at the buoy location is driven by the prevalence of the trade winds (Figure 6c). We find that faster (slower) monthly average wind speeds in 2023, relative to the long-term trends, follow a pattern of increased (decreased) frequency of occurrence of the trade winds (Figure 6b, c). For example, the average wind speed at the buoy location during February 2023 is the 96th percentile of the averages for all Februarys in the 40-year ERA5 record. The frequency of occurrence of the trade winds (50°-100°) during February 2023 (occurring 72% of the time) corresponds to a high percentile relative to all Februarys in the 40-year ERA5 record: 87th. Similarly, the slowest monthly wind speeds in 2023 relative to their 40-year counterparts (January: 9th percentile, March: 18th percentile, October: 4th percentile) have lower frequencies of trade wind occurrences during 2023 (January: 29th percentile, March: 1st percentile, October: 7th percentile) (Figure 6b, c)."

[Figure]

**Figure 6.** (a) Monthly average wind speeds from the Hawaii lidar observations, ERA5, and UH-WRF. (b) Normalized monthly average wind speeds from the Hawaii lidar observations, ERA5, and UH-WRF paired with ERA5 long-term normalized monthly wind speeds, which cover a 40-year period from 1985 to 2024. (c) Frequency of occurrence of winds sourcing between 50° and 100° from ERA5, both long-term and for 2023. Average observed and modelled wind speeds and trade wind frequencies for December reflect only the first half of the month, corresponding to the buoy deployment period.

We truly appreciate the approach you suggested to provide context for the seasonal variations in wind observed in 2023. Similarly, your suggestion provided us with an opportunity to explore the performance of ERA5 and UH-WRF during some interesting weather events and we are grateful for the resultant improvement to the analysis. As with the annual analysis, we concur with your guidance that we cannot perform helpful comparisons between the observations and NOW-23 and Global Wind Atlas given the uniqueness of the year 2023 and the incongruent temporal coverage periods, therefore we have removed discussion of the seasonal cycles of those two datasets.

CDS (Climate Data Store): ERA5 hourly data on single levels from 1940 to present [data set], https://cds.climate.copernicus.eu/datasets/reanalysis-era5-single-levels, accessed 26 November 2025b.

Longman, R. J., Elison Timm, O., Giambelluca, T. W., and Kaiser, L.: A 20-Year Analysis of Disturbance-Driven Rainfall on Oʻahu, Hawaiʻi, Monthly Weather Review, 149, 6, 1767-1783, https://doi.org/10.1175/MWR-D-20-0287.1, 2021.

Morrison, I. and Businger, S.: Synoptic Structure and Evolution of a Kona Low, Weather and Forecasting, 16, 81-98, https://doi.org/10.1175/1520-0434(2001)016%3C0081:SSAEOA%3E2.0.CO;2, 2001.

3. Local aspects

A great strength of this study is the fact that the wind is almost undisturbed tradewind flow. However, there may be some local influences relating to the effects of the islands. This could be especially relevant to the diurnal cycle in figure 13, and potentially particularly in the rare cases when the wind is blowing from the ESE or SSW. The lidar observations have a larger-amplitude diurnal cycle than either of the reanalysis datasets studied, but is this due to the different prevalence of wind directions in the different datasets? I suggest splitting the diurnal cycle up into different wind direction bins, although I know it will be difficult to get statistical significance for the rarer directions. A relevant reference for interaction of local phenomena with the tradewind environment is Dao et al. (2025):

Dao, T.L., Vincent, C.L., Huang, Y., Peatman, S.C., Soderholm, J.S., Birch, C.E., et al. (2025) Joint modulation of coastal rainfall in Northeast Australia by local and large-scale forcings. Quarterly Journal of the Royal Meteorological Society, e70027. Available from: https://doi.org/10.1002/qj.70027

Thank you for this important suggestion to explore the local effects of the islands! In both the original and the latest versions of the draft, we identified ERA5 as having the largest-amplitude diurnal cycle and attributed this finding to the discontinuities that appear in ERA5 which are tied to the start of the 12-hour reanalysis assimilation windows.

Lines 266-277: "The observed wind speeds transition smoothly across the diurnal cycle, with a standard deviation of the 140 m hourly average wind speeds of 0.28 m s$^{-1}$. The fastest speeds occur during the evening and at night, while the slowest speeds are observed during the day (Figure 7a, c). The UH-WRF diurnal cycle follows that of the observations, with nearly zero bias occurring for hours between 0 and 11 UTC and underestimation occurring between 12 and 23 UTC (average hourly bias = -0.40 m s$^{-1}$) (Figure 7b). Identifying necessary improvements to better characterize diurnal mixing within boundary layer parameterizations is required. Contrary to ERA5's accurate representation of the observed seasonal wind speed cycle, the reanalysis struggles to capture the observed diurnal cycle in 2023. Trends in the ERA5 wind speeds and associated wind speed biases are tied to the start of the 12-hour reanalysis assimilation windows of 9 and 21 UTC (Hersbach et al., 2020) as shown by the sharp peaks at these hours in Figure 7. Such discontinuities in the ERA5 diurnal wind speed cycle are also noted by Kalverla et al. (2019) over the North Sea. As with the annual and seasonal wind resource, 2023 is an atypical year relative to 40 years of ERA5 diurnal cycles (Figure 7d)."

[Figure]

**Figure 7.** (a) Hourly average wind speeds during 2023, (b) hourly wind speed bias during 2023, and (c) normalized hourly average wind speeds during 2023 from the Hawaii lidar observations, ERA5 and UH-WRF. (d) Normalized 2023 and long-term (1985-2024) ERA5 100 m wind speeds at the lidar buoy location.

To address your question on different prevalence of wind directions in the different datasets, we provide Lines 284-293 and Figure 8:

"The wind resource in Hawai'i is dominated by persistent northeast trade winds, with rare occurrences of wind sourcing from the south at O'ahu (Argüeso and Businger, 2018). During the overlapping period of the buoy observations and ERA5 and UH-WRF simulation coverage (1 January 2023 – 15 December 2023), 76% of the observed 140 m wind at the buoy location occur between 50° and 100° (Figure 8a). ERA5 underestimates the observed frequency of winds occurring between 50° and 100° at 70% (Figure 8b), while the percentage of winds occurring between 50° and 100° according to UH-WRF is closer to the observations at 73% (Figure 8c). Winds originating from O'ahu (230°-300°) (Figure 8a) constitute only 1% of the observed 2023 wind climatology at the lidar buoy location, suggesting that the buoy is situated within the wake of O'ahu island. Both ERA5 and UH-WRF slightly overestimate winds originating from the direction of O'ahu at 3% and 2%, respectively. Winds originating from Moloka'i (110°-150°) are slightly more frequent (3%) and ERA5 and UH-WRF similarly overestimate the rate of occurrence of Moloka'i-based winds at 5%."

[Figure]

**Figure 8.** Wind direction distributions at 140 m from (a) the O'ahu lidar buoy, (b) ERA5, and (c) UH-WRF between 1 January and 15 December 2023.

We also added an analysis of the dataset performance according to whether the winds were trade winds or less typical directions that could include influences from the islands on Lines 298-306:

"As expected given the trade wind-dominant environment, the median 140 m wind speed biases from ERA5 and UH-WRF are extremely similar whether considering the entire buoy deployment period (-1.41 m s⁻¹ for ERA5, -0.21 m s⁻¹ for UH-WRF) or just times when the trade winds are observed (-1.43 m s⁻¹ for ERA5, -0.22 m s⁻¹ for UH-WRF) (Figure 9). While keeping in mind that the sample sizes are significantly smaller when considering island-influenced winds, Figure 9 shows the wind speed biases becoming more positive for winds sourcing from the directions of Moloka'i and O'ahu. For winds originating from Moloka'i, the median ERA5 wind speed bias is -1.01 m s⁻¹, while UH-WRF exhibits a tendency to overestimate the wind speeds with a median bias of 0.46 m s⁻¹. In the rare event that winds at the lidar buoy location originated from O'ahu during 2023, the degree of simulation overestimation is notable, particularly when examining UH-WRF (median wind speed bias = 1.80 m s⁻¹)."

[Figure]

**Figure 9.** ERA5 and UH-WRF wind speed bias according to observed 140 m wind direction. The trade winds are defined as sourcing between 50° and 100°; Moloka'i-sourced winds are defined between 110° and 150°; O'ahu winds are defined between 230° and 300°.

Argüeso, D. and Businger, S.: Wind power characteristics off O'ahu, Hawaii, Renewable Energy, 128, 324-336, https://doi.org/10.1016/j.renene.2018.05.080, 2018.

Hersbach, H. and coauthors: The ERA5 Global Reanalysis, Q. J. Roy. Meteor. Soc., 146, 1999–2049, https://doi.org/10.1002/qj.3803, 2020.

Kalverla, P. C., Holtslag, A. A. M., Ronda, R. J., and Steeneveld, G-J.: Quality of wind characteristics in recent wind atlases over the North Sea, Quarterly Journal of the Royal Meteorological Society, 146(728), 1498-1515, https://doi.org/10.1002/qj.3748, 2020.

4. Treatment of the short dataset

As already mentioned before, the treatment of the short dataset is not quite convincing. The authors should use the ERA5 dataset to establish the links between the other datasets that all cover different periods: Eg. a plot of the monthly wind averages from ERA5 for the periods 1950-present, 2000-2019, 2008-2018 and 2023 will help tie

everything together, and clarify the representivity of the different time periods. This may indicate some further filtering as mentioned above, such as restricting the analysis to El Nino years.

We completely agree with your guidance regarding the short dataset. As mentioned in the response to Major Comment 1, we incorporated your suggestion to use the long-term ERA5 dataset to establish links between the other datasets. In short summary of our response to Major Comment 1, we found (thanks to your suggestion) that the year 2023 was such a unique wind year at the location of the lidar buoy deployment location that its characteristics would not be represented by any year within the temporal ranges of the nonconcurrent datasets NOW-23 and GWA3. Therefore, we opted to no longer speculate on any kind of performance relative to the 2023 observations for the nonconcurrent datasets.

Specific Comments

Lines 57-80. Could this information be presented in a more integrated way? Listing them one by one is difficult to read and doesn't leave any overall impression of the issues. Perhaps group by dataset type: "Several studies have evaluated ERA5 at offshore sites, and noted a low wind speed bias (ref ref). etc". I guess the key message is that there has only been a small number of studies so far, the results are inconsistent across the studies, which is what makes this current study important.

We have truncated and grouped Lines 53-65 as follows per your recommendation for improved readability:

"With the increasing availability (though still a relatively small sample) of offshore rotor level observations, studies have emerged over recent years comparing the performance of multiple simulation datasets at turbine rotor heights in offshore locations with the aims of aiding analysts in selecting the optimal datasets for resource assessments and highlighting areas for accuracy improvement for dataset developers. These studies have shown significant differences in performance across simulations being validated at a single offshore location, and in some cases one simulation product can be the best performer for one error metric (bias, correlation, mean absolute error, etc) and the worst performer for another metric. A variety of wind performance studies have evaluated ERA5 and note that the reanalysis tends to underestimate observed wind speeds (Kalverla et al., 2020 over the North Sea; Sheridan et al., 2020 off the eastern coast of the United States; Pronk et al., 2022 and Fragano and Colle, 2025 off the eastern coast of the United States; Sheridan et al., 2022 off the western coast of the United States). Several offshore studies find that higher-resolution datasets improve upon ERA5 in terms of wind speed bias (Kalverla et al., 2020 examining the Dutch Offshore Wind Atlas; Pronk et al., 2020 examining the Wind Integration National Dataset Toolkit Long-term Ensemble Dataset (WTK-LED); Fragano and Colle, 2025 examining NOW-23) but note that ERA5 outperforms the higher-resolution datasets according to other error metrics or wind field characteristics."

Line 92-92 – Define YSU and MYNN acronyms

The definitions of YSU and MYNN have been added to the specified lines (Lines 78 in the updated draft). We appreciate you pointing out that the definition of these terms later in the text is not helpful to readers of the Introduction.

Line 104 – is metocean actually a word?

We have replaced instances of "metocean" with "meteorological and oceanographic" or "atmospheric and oceanic" throughout the text.

Line 136 – "off of Oahu" -> "off Oahu"

The suggested grammatical edit has been incorporated into the latest draft (Line 537, now located in Appendix A). Thank you!

Figure 2 – Where does the 4 metre observation come from?

The 4 m observation comes from the surface met station that is aboard the buoy in addition to the lidar. Per your question, we have added the following context to Figure 2's caption to improve clarity and have added a citation to help readers access the publicly available data:

"Measurements at 60 m and higher source from the onboard lidar system (DOE, 2025a) while measurements at 4 m source from the onboard surface meteorological station (DOE, 2025b)."

*DOE (U.S. Department of Energy): 10 min Lidar Winds/Derived Data [data set], https://wdh.energy.gov/ds/buoy/lidar.z07.c0, accessed 6 January 2025a.*

*DOE (U.S. Department of Energy): Hawaii - Wind Sentinel (120), Oahu, Hawaii / Reviewed Data [data set], https://wdh.energy.gov/ds/buoy/buoy.z07.b0, accessed 6 January 2025b.*

Line 177 – 'EMCWF' -> 'ECMWF'

We have fixed this typo on Line 140 of the updated draft. Many thanks for bringing it to our attention!

Line 191 – Presumably the 1.5 km simulation didn't use a cumulus parametrisation?

Correct, the cumulative parameterization was run at for the regional domain but not the 1.5-km simulation. We have added clarification to Lines 143-147:

"The key configurations include the WRF Single-Moment 6-class Microphysics scheme for representing the cloud microphysical processes, the Betts-Miller-Janjic cumulus parameterization for convective processes on the regional domain, the Rapid Radiative Transfer Model for Global Circulation Models for longwave and shortwave radiation, the Noah land-surface model, and the YSU planetary boundary layer (PBL) scheme (Hsiao et al., 2020, 2021)."

*Hsiao, F., Chen, Y.-L., and Hitzl, D. E.: Heavy rainfall events over central O'ahu under weak wind conditions during seasonal transitions, Monthly Weather Review, 148, 4117-4141, https://doi.org/10.1175/MWR-D-19-0358.1, 2020.*

*Hsiao, F., Chen, Y.-L., Nguyen, H. P., Hitzl, D. E., and Ballard, R.: Effects of Trade Wind Strength on Airflow and Cloudiness over O'ahu, Monthly Weather Review, 149 (9), 3037-3062, https://doi.org/10.1175/MWR-D-20-0399.1, 2021.*

Table 1: The GWA3 dataset should mention the microscale downscaling, not just the mesoscale modelling part

We have removed analysis of GWA3 per the response to Major Comment 1. However, in appreciation of the reviewer's suggestion, we would have added the following to Table 1 for GWA3:

"WAsP IBZ model in the microscale phase", along with the accompanying text: "From there, microscale modelling using the linear IBZ flow models within the Wind Atlas Analysis and Application Program (WAsP) is used for horizontal and vertical extrapolation of the wind to achieve an output grid spacing of 250 m based on high-resolution orographic and roughness length maps (Davis et al., 2023)."

*Davis, N. N., Badger, J., Hahmann, A. N., Hansen, B. O., Mortensen, N. G., Kelly, M., Larsén, X. G., Olsen, B. T., Floors, R., Lizcano, G., Casso, P., Lacave, O., Bosch, A., Bauwens, I., Knight, O. J., Potter van Loon, A., Fox, R., Parvanyan, T., Krohn Hansen, S. B., Heathfield, D., Onninen, M., and Drummond, R.: The Global Wind Atlas: A High-Resolution Dataset of Climatologies and Associated Web-Based Application, Bulletin of the American Meteorological Society, 104.8, E1507-E1525, https://doi.org/10.1175/BAMS-D-21-0075.1, 2023.*

Equations 1-3. Are these standard equations really necessary? Perhaps check with the editors.

We have removed the equations for these commonly employed metrics per your suggestion.

Figure 3: It would be better to make the time period for these plots the same (1 Jan 2023 – 15 December 2023).

We appreciate your helpful suggestion for consistency and have updated the entirety of Section 3 to compare the performance of UH-WRF and ERA5 during the overlapping period when both datasets and the lidar observations are available (1 Jan 2023 – 15 Dec 2023 for hours when all three datasets are represented). Figure 3 and the associated discussion in the text now read (Lines 201-209):

"During the nearly year-long overlapping period of the buoy deployment, UH-WRF, and ERA5 (1 January 2023 – 15 December 2023), ERA5 strongly underestimates the observed wind speeds across the profile with biases ranging from -1.52 m s$^{-1}$ to -1.54 m s$^{-1}$ (-1.54 m s$^{-1}$ at 140 m) (Figure 4). UH-WRF underestimates the observed wind speeds across the profile as well, but to a lesser degree, with biases ranging from -0.24 m s$^{-1}$ to -0.29 m s$^{-1}$ (-0.25 m s$^{-1}$ at 140 m) (Figure 4). The correlations for both ERA5 (0.88-0.89) and UH-WRF (0.84-0.85) suggest successful representation of the hourly fluctuations in the observed wind speeds (Figure 4a). Previous DOE lidar buoy deployments off the northern and central coasts of California revealed similar correlations for ERA5 (0.88) (Sheridan et al., 2022). The CRMSEs during the Hawai'i buoy deployment (ERA5 = 1.59-1.62 m s$^{-1}$, UH-WRF = 1.96-2.02 m s$^{-1}$) (Figure 4a) are notably lower than the ERA5-based CRMSEs found during the California deployments, which ranged between 2.3 m s$^{-1}$ and 2.4 m s$^{-1}$."

[Figure]

**Figure 4.** ERA5 and UH-WRF wind speed (a) correlations, standard deviations, and CRMSEs, and (b), (c) distributions at the Hawaii buoy location during the overlapping period of 1 January 2023 – 15 December 2023.

Figure 4 and accompanying text: Define that you're talking about the median bias, not the mean bias.

We ensured that the accompanying text includes the word "median" when discussing the bias throughout Lines 221-227 and added the following clarification to the caption of Figure 5 (previously Figure 4): "For this and all box plots in the manuscript, the line in the centre of the box indicates the median."

"ERA5 and UH-WRF follow similar patterns in bias according to the observed wind speed (Figure 5). In representing the observed pre-cut-in wind speeds at 140 m, ERA5 exhibits little bias (median = 0.15 m s$^{-1}$) while UH-WRF significantly overestimates (median bias = 0.83 m s$^{-1}$). For observed wind speeds on the steep portion of the power curve, ERA5 and UH-WRF underestimate with median biases of -1.39 m s$^{-1}$ and -0.25 m s$^{-1}$. The greatest ERA5 errors are determined for observed wind speeds corresponding to maximum turbine generation, at the top of the power curve. For these faster observed wind speeds, ERA5 displays significant underestimation, with a median bias of -1.74 m s$^{-1}$, while UH-WRF produces a median bias of -0.30 m s$^{-1}$."

[Figure]

**Figure 5.** ERA5 and UH-WRF wind speed bias according to observed wind speed during the Hawaii buoy deployment. For this and all box plots in the manuscript, the line in the centre of the box indicates the median.

Line 284 – 'gently' -> 'smoothly'

We have swapped these words in the latest draft (Line 267), per your suggestion.

Figure 5 – perhaps a bar chart would be more appropriate, since this is not really continuous data.

With the switch to making the analysis of the 3 comparison datasets (observations, ERA5, and UH-WRF) concurrent (Specific Comment concerning Figure 3, above), the monthly analysis for Figure 5 (now Figure 6) is now continuous:

[Figure]

**Figure 6.** (a) Monthly average wind speeds from the Hawaii lidar observations, ERA5, and UH-WRF. (b) Monthly average wind speeds normalized by the annual average wind speed from the Hawaii lidar observations, ERA5, and UH-WRF paired with ERA5 long-term normalized wind speeds, which cover a 40-year period from 1985 to 2024. (c) Frequency of occurrence of winds sourcing between 50° and 100° from ERA5, both long-term and for 2023. Average observed and modelled wind speeds and trade wind frequencies for December reflect only the first half of the month, corresponding to the buoy deployment period.

Line 293. Clarify what these correlations are. Are they between the 24 values of the average diurnal cycle for each dataset?

Thank you for your helpful suggestion for clarity. Yes, the correlations were between the 24 hours of the diurnal cycle for each dataset. Upon review, we opted to remove this type of correlation analysis from the diurnal and seasonal discussions, as we felt that the high degree of correlation for such a small sample size obscured the true performance of the simulation datasets in representing the temporal trends. For example, if we examine the diurnal cycles, both ERA5 and UH-WRF provide correlations >0.9 when comparing with observations. However,

these high correlations obscure the clear challenges that ERA5 has with regard to representing the observed diurnal cycle, as seen in the scatter plots below that we include for the reviewer's benefit:

[Figure]

Figure 7 – I think it would be clearer to just show the observed wind rose.

We have simplified the figure (now Figure 8) to show the observed wind rose (Figure 8a) alongside the ERA5 and UH-WRF wind roses (Figure 8b,c):

[Figure]

**Figure 8.** Wind direction distributions at 140 m from (a) the O'ahu lidar buoy, (b) ERA5, and (c) UH-WRF between 1 January and 15 December 2023.

Figure 8 – It might be better to calculate the Richardson number or other stability metric, rather than just the airsea temperature difference.

We appreciate the suggestion that the air-sea temperature difference is not the ideal parameter to investigate, especially for a study focused on conditions at the wind turbine rotor level. We considered the relationship between atmospheric stability and the enhancement or suppression of turbulence and have reworked Section 3.3 and Figure 10 to instead analyze the ERA5 and UH-WRF performance according to turbulence intensity using the 1-Hz lidar buoy at a height of 140 m. This approach enables us to perform an observation-based evaluation at turbine hub height in lieu of a non-existent observed temperature profile or a parameterized stability metric such as the Obukhov length developed by, for example, the Coupled Ocean-Atmosphere Response Experiment (COARE) algorithm.

Lines 312-324: "Assessing the degree of turbulence at a location of wind energy development interest is advantageous for establishing generation expectations, particularly in a waked wind farm environment (Hansen et al., 2011). Therefore, it is important to define the baseline performance of datasets used for wind resource assessment according to different turbulent environments. Using the 1-Hz lidar buoy observations from the Hawai'i deployment (DOE, 2025c), we determine turbulence intensity (TI) at turbine hub height using the ratio of the standard deviation to the mean of the 140 m wind speeds over 10-minute periods (DOE, 2025a). At the offshore O'ahu buoy location, TI values below 0.1 occur 32% of the deployment period, while the bulk of the measurements (60%) show that TI during the deployment is between 0.1 and 0.2 (Figure 10a). The sample size of TI values reaching or exceeding 0.2 is small (8% of the deployment period). For hub height wind speeds below 5 m s$^{-1}$, the observed TI at the buoy location follows an inverse relationship with the observed mean 10-minute wind speeds and then hovers around 0.11 for wind speeds faster than 5 m s$^{-1}$ (Figure 10b). ERA5 and UH-WRF exhibit trends of increasingly positive wind speed bias with increasing hub height turbulence, with median ERA5 140 m wind speed biases of -1.63 m s$^{-1}$, -1.44 m s$^{-1}$, and -0.18 m s$^{-1}$ and median UH-WRF 140 m wind speed biases of -0.48 m s$^{-1}$, -0.16 m s$^{-1}$, and 0.75 m s$^{-1}$ for observed 140 m TI less than 0.1, between 0.1 and 0.2, and at least 0.2, respectively (Figure 10c, d)."

[Figure]

**Figure 10.** (a) Distribution of TI at 140 m during the Hawaii lidar buoy deployment, (b) observed TI according to observed 140 m wind speed, and (c) ERA5 and (d) UH-WRF 140 m wind speed bias according to observed TI.

*DOE (U.S. Department of Energy): 10 min Lidar Winds/Derived Data [data set], https://wdh.energy.gov/ds/buoy/lidar.z07.c0, accessed 6 January 2025a.*

*DOE (U.S. Department of Energy): 1Hz Lidar Winds / Reviewed Data [data set], https://wdh.energy.gov/ds/buoy/lidar.z07.b0, accessed 24 November 2025c.*

*Hansen, K. S., Barthelmie, R. J., Jensen, L. E., Sommer, A.: The impact of turbulence intensity and atmospheric stability on power deficits due to wind turbine wakes at Horns Rev wind farm, Wind Energy, 15(1), 183-196, https://doi.org/10.1002/we.512, 2011.*

Section 4.1. I'm really not sure that it's reasonable to talk about bias here, given the different time periods. See general comments. Use the same time period for ERA5, NOW23 and GWA3, and then also do it for just 2023 for ERA5.

We agree. Based on your suggestion to assess ERA5 for the NOW-23 and GWA3 time periods relative to 2023 and the long-term, we determined that it was unfeasible to perform any validation for NOW-23 and GWA3 using the 2023 lidar buoy observations off O'ahu. A full discussion of the investigation can be found in the response to your first Major Comment.

Figure 11 – What do the box plots show? Is it all values in the 2000-2019 period, or the distribution of the 20 annual averages during that time?

This figure has been removed from the latest draft to go, along with the analysis of NOW-23 and GWA3. For the reviewer, this figure showed the 20 annual averages from NOW-23 and the 10 annual averages from GWA3. We apologize for not making this clear in the caption.

Figure 13 - Why were the ERA5 and WRF datasets not included in the diurnal cycle study?

We apologize for not including them originally. If we were including NOW-23 and GWA3 in the latest draft, we would add ERA5 and WRF into the figure per your suggestion.

---

## Author Comment (AC2)

This study evaluates the performance of reanalysis, regional mesoscale simulations, and purpose built wind datasets in a promising yet sparsely observed offshore environment near Hawaii. Using a year-long lidar buoy deployment off the eastern coast of Oahu, it characterizes the local wind resource and validates four datasets (ERA5, UH-WRF, NOW-23, GWA3) at different spatial resolutions. The region's persistent trade winds make it favorable for wind power, but limited hub height observations underscore the need for accurate data. By comparing multiple products in this uniquely characterized setting, the work is relevant to the sector and appears suitable for publication pending the revisions below.

We are deeply grateful for your time and thoughtfulness in reviewing our manuscript and providing valuable suggestions for improvement. We are thankful for the opportunity to share an updated draft that reflects the guidance provided by yourself and an additional reviewer.

Prior to the responses to your specific concerns, which follow, we wished to share one major update to the manuscript that stemmed from analysis suggested by both reviewers: Based on the recommendation to evaluate whether the observation year 2023 was a typical or atypical wind resource year at the deployment location, we established that it was a highly atypical year that did not allow for proper evaluation of the nonconcurrent datasets NOW-23 and Global Wind Atlas. Therefore, we have removed them as analysis datasets from the manuscript with the following discussion:

Lines 180-194: "Prior to comparing the O'ahu observations, which predominantly occur in the year 2023, with atmospheric datasets, it is imperative to provide context on what kind of meteorological year 2023 is relative to the long-term interannual wind speed variability noted at the deployment location. Utilising annual averages of the Oceanic Niño Index (ONI), we find that 2023 is categorized as an El Niño year based on the Climate Prediction Center's threshold of +/- 0.5°C (Figure 3a) (NOAA, 2025). In Figure 3b, we explore the annual average ERA5 100 m wind speeds (CDS, 2025b) between 1985 and 2024 and determine that 2022 and 2023 are tied for having the lowest annual average wind speeds over the 40-year period (annual average wind speeds normalized by the 40-year mean for both years = 0.88) while being opposingly classified as La Niña and El Niño years, respectively (Figure 3a). According to ERA5, precipitation and 2 m temperature are above average at the buoy location for the year 2023 (Figure 3c, d). The above average precipitation, temperature, and weakening of the trade winds are consistent with expected El Niño characteristics (Lu et al., 2020).

The following analyses focus on the performance of two simulation datasets that have concurrent temporal coverage with the 2023 observations: ERA5 and UH-WRF. Given the context of 2023 being a record low wind resource year east of O'ahu according to the ERA5 record, it is important to note the need for long-term, continuously updating datasets like reanalyses. Purpose-built wind datasets, like NOW-23 (2000-2019) and Global Wind Atlas (2008-2017), provide numerous years for wind resource assessment but would not represent the characteristics of an atypical year like 2023."

[Figure]

**Figure 3.** (a) Annual average ONI, (b) normalized annual average 100 m wind speed from ERA5, (c) normalized annual total precipitation from ERA5, and (d) normalized annual average 2 m temperature from ERA5 over the 40-year period between 1985 and 2024 coloured by the annual average ONI. The annual average wind speeds are normalized by the 40-year average wind speed at the O'ahu buoy deployment location.

*CDS (Climate Data Store): ERA5 hourly data on single levels from 1940 to present [data set], https://cds.climate.copernicus.eu/datasets/reanalysis-era5-single-levels, accessed 26 November 2025b.*

*Lu, B-Y., Chu, P-S., Kim, S-H., and Karamperidou, C.: Hawaiian Regional Climate Variability during Two Types of El Niño, Journal of Climate, 33, 22, 9929-9943, https://doi.org/10.1175/JCLI-D-19-0985.1, 2020.*

*NOAA (National Oceanic and Atmospheric Administration): Climate Variability: Oceanic Niño Index, https://www.climate.gov/news-features/understanding-climate/climate-variability-oceanic-nino-index, last updated 25 June 2025.*

Major concerns

1. Attribution to PBL choice

The manuscript notes differences in PBL schemes across datasets but does not isolate the PBL contribution with, for instance, a sensitivity test. The study even presents several diagnostics informative of boundary-layer processes (e.g., stability-stratified errors, diurnal cycle, vertical shear), but the connection from these results to PBL-scheme attribution is not made explicit. If this is the case, please state explicitly how each diagnostic reflects expected PBL behavior, or soften the claim and note that other factors are in play.

We are grateful for your feedback on this matter and agree that the language in the original draft was too strong regarding differences in PBL schemes in lieu of actual sensitivity testing. With respect to your concern, we have softened the PBL attribution language and diversified the discussion to include other factors as follows:

Lines 14-17:

We left this sentence as is: "The degree of such errors is influenced by a number of factors, including spatial resolution and the handling of processes within the planetary boundary layer (PBL)."

But deleted some of the language in the following sentence: "In this work, we provide a wind resource characterization from year-long lidar buoy measurements off the eastern coast of O'ahu, Hawaii, an environment previously unobserved at the rotor level, and use it to establish the performance of two simulation datasets ."

In the Introduction, simulation error is attributed to a number of possible causes, including choice of PBL scheme but also spatial resolution, ability to resolve small scale features, ability to resolve coastal influences, and ability to represent the diurnal wind speed cycle on Lines 66-80:

"Variations across wind simulations for performance metrics like bias and correlation occur for a number of reasons. In their evaluation of reanalysis products, Ramon et al. (2019) found that the lowest correlations for wind speeds compared with global tall tower observations corresponded to the coarsest resolution grids. Similarly, Kalverla et al. (2020) attributed ERA5's horizontal resolution to underestimation of observed offshore wind ramps due to limitations in the model representation of the small-scale structures responsible for ramps. Sheridan et al. (2022) noted that the high correlation of the Rapid Refresh model with offshore California observations was at least partly due to the model's higher resolution and therefore better ability at resolving coastal features and phenomena that coarser models miss. Pronk et al. (2022) determined that the preliminary WRF simulations for WTK-LED outperformed ERA5 in terms of bias at an onshore site and an offshore site in the United States but found the opposite behaviour for centred root-mean-square error (CRMSE). Pronk et al. (2022) suspected that the underperformance of WTK-LED for CRMSE is due to WTK-LED's exaggeration of the diurnal cycle of wind speeds at both study sites, especially the onshore location. Bodini et al. (2024b) tested two PBL schemes in simulations off the coast of California using lidar buoy observations and established that using the Mellor-Yamada-Nakanishi-Niino (MYNN) scheme overestimated stability compared with observations and simulations using the Yonsei University (YSU) scheme, resulting in overestimation of offshore wind speeds and the selection of YSU as the PBL scheme for the NOW-23 South Pacific region."

Deletion on Lines 502-504:

"UH-WRF exhibits a significantly smaller bias (-0.25 m s$^{-1}$) than ERA5 and a correlation (0.85) close to ERA5 (0.89) at the O'ahu site."

2. Shifting focus from validation to campaign report

The buoy campaign is essential for the validation, but exhaustive operational/installation details distract from the paper's focus. Keep only what is needed for readers to understand, reproduce, and trust the results (site, period, completeness, QC, uncertainties, key post processing). Technical but relevant info can go in the Appendix, but non-essential logistics/engineering details should be moved to a separate deployment report.

We appreciate this helpful suggestion to not distract from the validation with extensive deployment information. Per your recommendation, we have significantly truncated Section 2.1 to the following, with guidance to the reader to seek more information in the Appendix if desired:

Lines 110-119:

"**2.1 Buoy observations**

DOE owns multiple AXYS WindSentinel™ buoys, including the buoy deployed for resource assessment off the coast of O'ahu which is outfitted with a Leosphere Windcube v2 lidar system and surface meteorological and oceanographic instruments (Figure 1). While the DOE buoy observations during the Hawai'i deployment are discussed extensively in Appendix A, we provide the key characteristics of the lidar wind measurements to set a baseline for comparison with the wind datasets. Many of the observed characteristics are also found throughout the results to provide context for the evaluations of wind dataset performance. The DOE buoy was deployed approximately 25 km off the coast of O'ahu between 1 December 2022 and 15 December 2023 (DOE, 2025a, b, c) (Figure 1). This study focuses on the wind data measured by the lidar aboard the buoy at every 20 m between 60 m and 240 m. Following the quality control performed as discussed in Appendix A, lidar data recovery for the deployment period is 98% or higher for all heights between 60 m and 240 m."

*DOE (U.S. Department of Energy): 10 min Lidar Winds/Derived Data [data set], https://wdh.energy.gov/ds/buoy/lidar.z07.c0, accessed 6 January 2025a.*

*DOE (U.S. Department of Energy): Hawaii - Wind Sentinel (120), Oahu, Hawaii / Reviewed Data [data set], https://wdh.energy.gov/ds/buoy/buoy.z07.b0, accessed 6 January 2025b.*

*DOE (U.S. Department of Energy): 1Hz Lidar Winds / Reviewed Data [data set], https://wdh.energy.gov/ds/buoy/lidar.z07.b0, accessed 24 November 2025c.*

Minor revisions

L89: Several times in the text, the word "trend(s)" could be replaced with "result(s)", "pattern(s)" or "behavior".

We appreciate this caution against overuse of the word "trend" and have diversified with your recommended words throughout the text.

L92: Define MYNN and YSU on first use.

The definitions of YSU and MYNN have been added to the specified lines. We appreciate you pointing out that the definition of these terms later in the text is not helpful to readers of the Introduction.

Lines 76-80: "Bodini et al. (2024b) tested two PBL schemes in simulations off the coast of California using lidar buoy observations and established that using the Mellor-Yamada-Nakanishi-Niino (MYNN) scheme overestimated stability compared with observations and simulations using the Yonsei University (YSU) scheme, resulting in overestimation of offshore wind speeds and the selection of YSU as the PBL scheme for the NOW-23 South Pacific region."

*Bodini, N., Optis, M., Liu, M. Gaudet, B. Krishnamurthy, R., Kumler, A., Rosencrans, D., Rybchuk, A., Tai, S.-L., Berg, L., Musial, W., Lundquist, J. K., Purkayastha, A., Young, A., and Draxl, C.: Causes of and Solutions to Wind Speed Bias in NREL's 2020 Offshore Wind Resource Assessment for the California Pacific Outer Continental Shelf, National Renewable Energy Laboratory (NREL), Golden, Colorado, United States, NREL/TP-5000-88215, https://docs.nrel.gov/docs/fy24osti/88215.pdf, 2024b.*

L93: Remove "using MYNN".

We have removed "using MYNN" per your recommendation (see excerpt in the response to the previous comment).

L97: "Nunalee and Basu (2014) noted …" — consider removing if it does not add to the argument in that paragraph.

We have removed this reference and thank you for the suggestion.

L101: Is there a reference for the observational campaign (e.g., Krishnamurthy et al., 2023) and/or others; point to Appendix A for site-specific details.

No specific reference for this particular observational campaign exists; therefore, we aim to provide the relevant details for the deployment in Appendix A (field summary, data recovery, post-processing, and quality-control). We have added the reference to Appendix A on Lines 84-85 per your recommendation:

"In late 2022, one of the lidar buoys was deployed off the eastern shore of O'ahu to gather meteorological and oceanographic observations (Appendix A)."

L103: Replace "hitherto have not been explored in an observational manner" with "had not previously been observed."

We have reworded the sentence according to your recommended phrasing.

Lines 81-84: "To address the need for wind resource characterization and model validation in offshore environments, the U.S. Department of Energy (DOE) has collaborated with the Bureau of Ocean Energy Management (BOEM) to deploy multiple buoy-mounted research lidars in locations that had not previously been observed at relevant heights for deep water energy development."

L114: "…ERA5 will exhibit a low wind speed bias …" this entire sentence is ambiguous.

Thank you. We have removed this sentence per your guidance.

Section 2: Consider merging the opening of Section 2 with the first paragraph of Section 2.1 to present a concise summary of the observational data (source, location, period, sampling/averaging, heights, instrument/method). The rest could be moved to a new subsection (e.g., Local Wind Characterization). You could briefly summarize the key points and refer to a single comprehensive figure that combines the already shown plots 2a and 2b, and adds (c) monthly and seasonal means and (d) the diurnal cycle, rather than exhaustively listing numbers. Any additional technical details should go in the Appendix.

We appreciate your recommendation to truncate the amount of detail in Section 2. We have taken your suggestions to merge the opening of Section 2 with the first paragraph of Section 2.1, establish a new subsection called Local Wind Characterization with a single comprehensive figure, and remove extensively listed numbers as follows:

Lines 106-130:

"**2 Data and methodology**

To set the stage for assessing the performance of reanalysis and mesoscale models using lidar buoy observations in a new offshore location, the next sections present an overview of the buoy observations, a brief characterization of the local wind resource, descriptions of the models that will be validated, and the procedure for establishing model performance success.

**2.1 Buoy observations**

DOE owns multiple AXYS WindSentinel™ buoys, including the buoy deployed for resource assessment off the coast of O'ahu which is outfitted with a Leosphere Windcube v2 lidar system and surface meteorological and oceanographic instruments (Figure 1). While the DOE buoy observations during the Hawai'i deployment are discussed extensively in Appendix A, we provide the key characteristics of the lidar wind measurements to set a baseline for comparison with the wind datasets. Many of the observed characteristics are also found throughout the results to provide context for the evaluations of wind dataset performance. The DOE buoy was deployed approximately 25 km off the coast of O'ahu between 1 December 2022 and 15 December 2023 (DOE, 2025a, b, c) (Figure 1). This study focuses on the wind data measured by the lidar aboard the buoy at every 20 m between 60 m and 240 m. Following the quality control performed as discussed in Appendix A, lidar data recovery for the deployment period is 98% or higher for all heights between 60 m and 240 m.

[Figure]

**Figure 1.** (a) Map of location of the Hawaii DOE lidar buoy deployment (indicated by the star). (b) Photo of one of the DOE lidar buoys by Ocean Tech Services, LLC and Pacific Northwest National Laboratory.

**2.2 Local wind characterization**

For the overlapping period covering the buoy deployment, ERA5, and UH-WRF (1 January – 15 December 2023), annual average wind speeds range from 8.87 m s⁻¹ at 60 m to 9.34 m s⁻¹ at 240 m with very little shear across the profile (Figure 2a). The observed winds predominantly source from the east-northeast with very little veer across the profile (Figure 2b). The observed 2023 seasonal and diurnal variations in the lidar wind speeds are presented in detail in Section 3.1, but briefly, significant variation is noted for the month-by-month transitions (Figure 2c) while the diurnal wind speed cycle transitions smoothly from hour-to-hour (Figure 2d)."

[Figure]

**Figure 2.** Observed (a) wind speed, (b) wind direction, (c) monthly average wind speed, and (d) hourly average wind speed by height a.s.l. from the Hawaii lidar buoy during 1 January 2023 – 15 December 2023. Measurements at 60 m and higher source from the onboard lidar system (DOE, 2025a) while measurements at 4 m source from the onboard surface meteorological station (DOE, 2025b).

Section 2.2: Clarify whether the ERA5 time series was horizontally interpolated or taken from the nearest sea or land/sea grid cell. The proximity to the coastline could influence results.

Thank you for pointing out this missing information. We have added the following text to Lines 170-176 to provide clarity based on your helpful recommendation:

"The wind datasets selected for evaluation range widely in terms of horizontal spatial resolution, from 1.5-km (UH-WRF) to approximately 27-km at the location of the buoy deployment (ERA5) (Table 1). For each dataset, we perform inverse distance weighting on the atmospheric variables to localize them to the location of the lidar buoy. For reference, the nearest grid points to the buoy location are at distances of 0.9 km (UH-WRF) and 10 km (ERA5). The four surrounding UH-WRF grid points to the buoy are at distances of 21-24 km from the nearest coastline (O'ahu). The four surrounding ERA5 grid points to the buoy are at distances of 3 km (Moloka'i), 17 km (O'ahu), 24 km (O'ahu), and 31 km (Moloka'i) from the nearest coastline."

L179: Provide a reference for UH-WRF if available.

A manuscript about UH-WRF is currently under development by the University of Hawaii team. In lieu of that, we cite previous simulations with similar model setups from the University of Hawaii team on Lines 143-147:

"The key configurations include the WRF Single-Moment 6-class Microphysics scheme for representing the cloud microphysical processes, the Betts-Miller-Janjic cumulus parameterization for convective processes on the regional domain, the Rapid Radiative Transfer Model for Global Circulation Models for longwave and shortwave radiation, the Noah land-surface model, and the YSU planetary boundary layer (PBL) scheme (Hsiao et al., 2020, 2021)."

*Hsiao, F., Chen, Y.-L., and Hitzl, D. E.: Heavy rainfall events over central O'ahu under weak wind conditions during seasonal transitions, Monthly Weather Review, 148, 4117-4141, https://doi.org/10.1175/MWR-D-19-0358.1, 2020.*

*Hsiao, F., Chen, Y.-L., Nguyen, H. P., Hitzl, D. E., and Ballard, R.: Effects of Trade Wind Strength on Airflow and Cloudiness over O'ahu, Monthly Weather Review, 149 (9), 3037-3062, https://doi.org/10.1175/MWR-D-20-0399.1, 2021.*

L205: Clarify that GWA3 wind data are provided as annual, monthly, and diurnal climatologies, not continuous hourly time series.

This sentence was removed along with the analysis of GWA3 and NOW-23 due to their nonconcurrence with the observations during an atypical wind resource year.

Table 1: Provide actual vertical levels used rather than "lowest N model heights." For ERA5, "PBL scheme plus the effect of data assimilation"?

We have updated Table 1 per your recommendations:

**Table 1.** Characteristics of wind assessment products evaluated in this analysis.

| Product | ERA5 | UH-WRF |
|---|---|---|
| **Type** | Reanalysis | Meteorological dataset |
| **Developers** | ECMWF | University of Hawaii |
| **Temporal coverage** | 1950 – present | 2023 |
| **Temporal resolution** | 1-hr | 1-hr |
| **Spatial coverage** | Global | Hawaii |
| **Spatial resolution** | 31-km | 1.5-km |
| **PBL handling** | First-order K-diffusion closure of Monin-Obukhov similarity theory in the surface layer and above the surface layer, except for unstable conditions when scheme is an eddy-diffusivity mass flux framework (Fragano and Colle, 2025) plus effects of data assimilation | YSU scheme |
| **Wind output heights used in this study** | Lowest 9 model heights: 4 m, 26 m, 50 m, 76 m, 105 m, 136 m, 171 m, 208 m, 249 m at the buoy location | Lowest 6 model heights: 0 m, 51 m, 85 m, 145 m, 187 m, 256 m at the buoy location |

*Fragano, C. G. and Colle, B. A.: Validation of Offshore Winds in the ERA5 Reanalysis and NREL NOW-23 WRF Analysis Using Two Floating Lidars in the New York Bight, Weather and Forecasting, 1307-1323, https://doi.org/10.1175/WAF-D-24-0155.1, 2025.*

L238–239: It appears the manuscript uses "average/median bias" when referring to mean/median errors. It sounds a bit redundant as bias is already the mean error. Unless you compute multiple bias values (e.g., per height or per wind speed range) and then take their mean/median. Remove the word "average" in both lines if redundant and check the entire manuscript.

Thank you. We have removed the redundant instances of "average" in the sentences in question and have scanned the remainder of the manuscript to ensure that such phrasing does not appear elsewhere.

Lines 201-204: "During the nearly year-long overlapping period of the buoy deployment, UH-WRF, and ERA5 (1 January 2023 – 15 December 2023), ERA5 strongly underestimates the observed wind speeds across the profile with biases ranging from -1.52 m s$^{-1}$ to -1.54 m s$^{-1}$ (-1.54 m s$^{-1}$ at 140 m) (Figure 4). UH-WRF underestimates the observed wind speeds across the profile as well, but to a lesser degree, with biases ranging from -0.24 m s$^{-1}$ to -0.29 m s$^{-1}$ (-0.25 m s$^{-1}$ at 140 m) (Figure 4)."

L241–243: "Figure 3a".

We have updated the references on Lines 204-209 of the updated manuscript to "Figure 4a", reflecting the figure numbering in the updated manuscript.

"The correlations for both ERA5 (0.88-0.89) and UH-WRF (0.84-0.85) suggest successful representation of the hourly fluctuations in the observed wind speeds (Figure 4a). Previous DOE lidar buoy deployments off the northern and central coasts of California revealed similar correlations for ERA5 (0.88) (Sheridan et al., 2022). The CRMSEs during the Hawai'i buoy deployment (ERA5 = 1.59-1.62 m s⁻¹, UH-WRF = 1.96-2.02 m s⁻¹) (Figure 4a) are notably lower than the ERA5-based CRMSEs found during the California deployments, which ranged between 2.3 m s⁻¹ and 2.4 m s⁻¹."

*Sheridan, L. M., Krishnamurthy, R., García Medina, G., Gaudet, B. J., Gustafson Jr., W. I., Mahon, A. M., Shaw, W. J., Newsom, R. K., Pekour, M., and Yang, Z.: Offshore reanalysis wind speed assessment across the wind turbine rotor layer off the United States Pacific coast, Wind Energy Science, 7, 2059-2084, https://doi.org/10.5194/wes-7-2059-2022, 2022.*

Figure 3: There is no plot 3d.

Thank you! We have removed "(d)" from the caption.

[Figure]

**Figure 3.** ERA5 and UH-WRF wind speed (a) correlations, standard deviations, and CRMSEs, and (b), (c) distributions at the Hawaii buoy location during the overlapping period of 1 January 2023 – 15 December 2023.

Fig. 5: State normalized by what and its relevance in the text.

We have updated the caption to read "Monthly average wind speeds normalized by the annual average wind speed" and Lines 236-239 as follows:

"Looking at the monthly wind speeds normalized by the annual average wind speed from the 2023 deployment observations and the simulation datasets, which show similar patterns, in the context of 40 years (1985 to 2024) of normalized monthly 100 m wind speeds from ERA5 (CDS, 2025b), the 2023 seasonal wind speed cycle is atypical relative to the long-term (Figure 6b)."

[Figure]

**Figure 6.** (a) Monthly average wind speeds from the Hawaii lidar observations, ERA5, and UH-WRF. (b) Monthly average wind speeds normalized by the annual average wind speed from the Hawaii lidar observations, ERA5, and UH-WRF paired with ERA5 long-term normalized monthly wind speeds, which cover a 40-year period from 1985 to 2024. (c) Frequency of occurrence of winds sourcing between 50° and 100° from ERA5, both long-term and for 2023. Average observed and modelled wind speeds and trade wind frequencies for December reflect only the first half of the month, corresponding to the buoy deployment period.

*CDS (Climate Data Store): ERA5 hourly data on single levels from 1940 to present [data set], https://cds.climate.copernicus.eu/datasets/reanalysis-era5-single-levels, accessed 26 November 2025b.*

Section 3.2: Add one sentence explaining why testing different atmospheric stability conditions is relevant. Also, state explicitly that you use air-sea temperature differential as a proxy for stability and note the limitations.

*Both reviewers had concerns about the air-sea temperature differential as a proxy for stability; therefore, we reworked Section 3.2 (3.3 in the updated manuscript) by replacing the air-sea temperature analysis with a discussion on the observed turbulence intensity at the lidar buoy location and the performance of the models according to TI. We hope that this analysis, using the 1-Hz lidar observations at hub height, is a more useful and*

acceptable analysis than exploring wind speed performance at hub height relative to parameters at the surface. Lines 311-324:

**3.3 Wind dataset performance according to turbulence intensity**

Assessing the degree of turbulence at a location of wind energy development interest is advantageous for establishing generation expectations, particularly in a waked wind farm environment (Hansen et al., 2011). Therefore, it is important to define the baseline performance of datasets used for wind resource assessment according to different turbulent environments. Using the 1-Hz lidar buoy observations from the Hawai'i deployment (DOE, 2025c), we determine turbulence intensity (TI) at turbine hub height using the ratio of the standard deviation to the mean of the 140 m wind speeds over 10-minute periods (DOE, 2025a). At the offshore O'ahu buoy location, TI values below 0.1 occur 32% of the deployment period, while the bulk of the measurements (60%) show that TI during the deployment is between 0.1 and 0.2 (Figure 10a). The sample size of TI values reaching or exceeding 0.2 is small (8% of the deployment period). For hub height wind speeds below 5 m s$^{-1}$, the observed TI at the buoy location follows an inverse relationship with the observed mean 10-minute wind speeds and then hovers around 0.11 for wind speeds faster than 5 m s$^{-1}$ (Figure 10b). ERA5 and UH-WRF exhibit trends of increasingly positive wind speed bias with increasing hub height turbulence, with median ERA5 140 m wind speed biases of -1.63 m s$^{-1}$, -1.44 m s$^{-1}$, and -0.18 m s$^{-1}$ and median UH-WRF 140 m wind speed biases of -0.48 m s$^{-1}$, -0.16 m s$^{-1}$, and 0.75 m s$^{-1}$ for observed 140 m TI less than 0.1, between 0.1 and 0.2, and at least 0.2, respectively (Figure 10c, d)."

[Figure]

**Figure 10.** (a) Distribution of TI at 140 m during the Hawaii lidar buoy deployment, (b) observed TI according to observed 140 m wind speed, and (c) ERA5 and (d) UH-WRF 140 m wind speed bias according to observed TI.

*DOE (U.S. Department of Energy): 10 min Lidar Winds/Derived Data [data set], https://wdh.energy.gov/ds/buoy/lidar.z07.c0, accessed 6 January 2025a.*

*DOE (U.S. Department of Energy): 1Hz Lidar Winds / Reviewed Data [data set], https://wdh.energy.gov/ds/buoy/lidar.z07.b0, accessed 24 November 2025c.*

*Hansen, K. S., Barthelmie, R. J., Jensen, L. E., Sommer, A.: The impact of turbulence intensity and atmospheric stability on power deficits due to wind turbine wakes at Horns Rev wind farm, Wind Energy, 15(1), 183-196, https://doi.org/10.1002/we.512, 2011.*

L358: Include results/discussion on occurrences of LLJs, or remove earlier references if not analyzed.

We have removed references to LLJs, as their occurrence was not notable in the O'ahu lidar buoy observations.

L368: Reference figure 11a.

This section and figure were removed along with the analysis of GWA3 and NOW-23 due to their nonconcurrence with the observations during an atypical wind resource year.

L383: Specify inconsistencies relative to what.

This sentence was removed along with the analysis of GWA3 and NOW-23 due to their nonconcurrence with the observations during an atypical wind resource year.

L433: Use "corroborate previous reports of" instead of "add another geographic data point to the trend of".

We have reworded the sentence per your recommended phrasing.

Lines 479-482: "In particular, the results of this investigation near Hawai'i corroborate previous reports of ERA5's underestimation of observed marine boundary layer winds as documented by Kalverla et al. (2020) over the North Sea, Sheridan et al. (2020) off the coasts of New Jersey and Virginia, United States, Pronk et al. (2022) and Fragano and Colle (2025) off the coast of New Jersey, United States, and Sheridan et al. (2022) off the coast of central California, United States."

*Fragano, C. G. and Colle, B. A.: Validation of Offshore Winds in the ERA5 Reanalysis and NREL NOW-23 WRF Analysis Using Two Floating Lidars in the New York Bight, Weather and Forecasting, 1307-1323, https://doi.org/10.1175/WAF-D-24-0155.1, 2025.*

*Kalverla, P. C., Holtslag, A. A. M., Ronda, R. J., and Steeneveld, G-J.: Quality of wind characteristics in recent wind atlases over the North Sea, Quarterly Journal of the Royal Meteorological Society, 146(728), 1498-1515, https://doi.org/10.1002/qj.3748, 2020.*

*Pronk, V., Bodini, N., Optis, M., Lundquist, J. K., Moriarty, P., Draxl, C., Purkayastha, A., and Young, E.: Can reanalysis products outperform mesoscale numerical weather prediction models in modeling the wind resource in simple terrain?, Wind Energ. Sci., 7, 487–504, https://doi.org/10.5194/wes-7-487-2022, 2022.*

*Sheridan, L. M., Krishnamurthy, R., Gorton, A. M., Shaw, W. J., and Newsom, R. K.: Validation of Reanalysis-Based Offshore Wind Resource Characterization Using Lidar Buoy Observation, Mar. Technol. Soc. J., 54, 44–61, https://doi.org/10.4031/MTSJ.54.6.13, 2020.*

*Sheridan, L. M., Krishnamurthy, R., García Medina, G., Gaudet, B. J., Gustafson Jr., W. I., Mahon, A. M., Shaw, W. J., Newsom, R. K., Pekour, M., and Yang, Z.: Offshore reanalysis wind speed assessment across the wind turbine rotor layer off the United States Pacific coast, Wind Energy Science, 7, 2059-2084, https://doi.org/10.5194/wes-7-2059-2022, 2022.*

L437–438: The winds may be consistent, but was 2023 a regular year? Maybe it would be helpful to include this information. Also, what is the distance from the buoy to the nearest coastline? If the ERA5 grid cell includes or lies close to the coast, this proximity could influence the results.

We have added the following details to satisfy your request for information on the distances from the coast:

Lines 116-117: "The DOE buoy was deployed approximately 25 km off the coast of O'ahu between 1 December 2022 and 15 December 2023."

Lines 173-176: "For reference, the nearest grid points to the buoy location are at distances of 0.9 km (UH-WRF) and 10 km (ERA5). The four surrounding UH-WRF grid points to the buoy are at distances of 21-24 km from the nearest coastline (O'ahu). The four surrounding ERA5 grid points to the buoy are at distances of 3 km (Moloka'i), 17 km (O'ahu), 24 km (O'ahu), and 31 km (Moloka'i) from the nearest coastline."

As for whether 2023 was a regular year, both you and Reviewer 1 asked this important question, which refocused the analysis of this manuscript to validating only the concurrent simulations, ERA5 and UH-WRF. Your suggested evaluation revealed that 2023 was not a typical year for wind resource off the east coast of O'ahu. We have added the following context to Lines 180-194 per your helpful recommendation:

"Prior to comparing the O'ahu observations, which predominantly occur in the year 2023, with atmospheric datasets, it is imperative to provide context on what kind of meteorological year 2023 is relative to the long-term interannual wind speed variability noted at the deployment location. Utilising annual averages of the Oceanic Niño Index (ONI), we find that 2023 is categorized as an El Niño year based on the Climate Prediction Center's threshold of +/- 0.5°C (Figure 3a) (NOAA, 2025). In Figure 3b, we explore the annual average ERA5 100 m wind speeds (CDS, 2025b) between 1985 and 2024 and determine that 2022 and 2023 are tied for having the lowest annual average wind speeds over the 40-year period (annual average wind speeds normalized by the 40-year mean for both years = 0.88) while being opposingly classified as La Niña and El Niño years, respectively (Figure 3a). According to ERA5, precipitation and 2 m temperature are above average at the buoy location for the year 2023 (Figure 3c, d). The above average precipitation, temperature, and weakening of the trade winds are consistent with expected El Niño characteristics (Lu et al., 2020).

The following analyses focus on the performance of two simulation datasets that have concurrent temporal coverage with the 2023 observations: ERA5 and UH-WRF. Given the context of 2023 being a record low wind resource year east of O'ahu according to the ERA5 record, it is important to note the need for long-term, continuously updating datasets like reanalyses. Purpose-built wind datasets, like NOW-23 (2000-2019) and Global Wind Atlas (2008-2017), provide numerous years for wind resource assessment but would not represent the characteristics of an atypical year like 2023."

[Figure]

**Figure 3.** (a) Annual average ONI, (b) normalized annual average 100 m wind speed from ERA5, (c) normalized annual total precipitation from ERA5, and (d) normalized annual average 2 m temperature from ERA5 over the 40-year period between 1985 and 2024 coloured by the annual average ONI. The annual average wind speeds are normalized by the 40-year average wind speed at the O'ahu buoy deployment location.

The 2023 seasonal and diurnal wind speed variations were also atypical relative to 40 years of ERA5 data, and we have included the following text and figures to highlight this determination.

Lines 236-249: "Looking at the monthly wind speeds normalized by the annual average wind speed from the 2023 deployment observations and the simulation datasets, which show similar patterns, in the context of 40 years (1985 to 2024) of normalized monthly 100 m wind speeds from ERA5 (CDS, 2025b), the 2023 seasonal wind speed cycle is atypical relative to the long-term (Figure 6b). While weather events can occur along the eastern shores of O'ahu that alter the typical conditions (Morrison and Businger, 2001) and potentially the temporal wind speed patterns (see Section 3.7), long-term ERA5-based analysis indicates that the 2023 seasonal wind speed cycle at the buoy location is driven by the prevalence of the trade winds (Figure 6c). We find that faster (slower) monthly average wind speeds in 2023, relative to the long-term trends, follow a pattern of increased (decreased) frequency of occurrence of the trade winds (Figure 6b, c). For example, the average wind speed at the buoy location during February 2023 is the 96th percentile of the averages for all Februarys in the 40-year ERA5 record. The frequency of occurrence of the trade winds (50°-100°) during February 2023 (occurring 72% of the time) corresponds to a high percentile relative to all Februarys in the 40-year ERA5 record: 87th. Similarly, the slowest monthly wind speeds in 2023 relative to their 40-year counterparts (January: 9th percentile, March: 18th percentile, October: 4th percentile) have lower frequencies of trade wind occurrences during 2023 (January: 29th percentile, March: 1st percentile, October: 7th percentile) (Figure 6b, c)."

[Figure]

**Figure 6.** (a) Monthly average wind speeds from the Hawai'i lidar observations, ERA5, and UH-WRF. (b) Monthly average wind speeds normalized by the annual average wind speed from the Hawai'i lidar observations, ERA5, and UH-WRF paired with ERA5 long-term normalized monthly wind speeds, which cover a 40-year period from 1985 to 2024. (c) Frequency of occurrence of winds sourcing between 50° and 100° from ERA5, both long-term and for 2023. Average observed and modelled wind speeds and trade wind frequencies for December reflect only the first half of the month, corresponding to the buoy deployment period.

Lines 276-277: "As with the annual and seasonal wind resource, 2023 is an atypical year relative to 40 years of ERA5 diurnal cycles (Figure 7d)."

[Figure]

**Figure 7.** (a) Hourly average wind speeds during 2023, (b) hourly wind speed bias during 2023, and (c) normalized hourly average wind speeds during 2023 from the Hawai'i lidar observations, ERA5 and UH-WRF. (d) Normalized 2023 and long-term (1985-2024) ERA5 100 m wind speeds at the lidar buoy location.

*CDS (Climate Data Store): ERA5 hourly data on single levels from 1940 to present [data set], https://cds.climate.copernicus.eu/datasets/reanalysis-era5-single-levels, accessed 26 November 2025b.*

*Morrison, I. and Businger, S.: Synoptic Structure and Evolution of a Kona Low, Weather and Forecasting, 16, 81-98, https://doi.org/10.1175/1520-0434(2001)016%3C0081:SSAEOA%3E2.0.CO;2, 2001.*

*NOAA (National Oceanic and Atmospheric Administration): Climate Variability: Oceanic Niño Index, https://www.climate.gov/news-features/understanding-climate/climate-variability-oceanic-nino-index, last updated 25 June 2025.*

L453–455: Justify attributing UH-WRF's smaller bias primarily to the PBL scheme (YSU) rather than to other differences.

As mentioned in the response to Major Comment 2, we have softened the claim regarding PBL influences due to the lack of sensitivity tests available at the observational location. The sentence in question is now worded as follows:

Lines 502-504: "UH-WRF exhibits a significantly smaller bias (-0.25 m s⁻¹) than ERA5 and a correlation (0.85) close to ERA5 (0.89) at the O'ahu site."

L487: There is no Shaw et al., 2020 in the references. Do you mean Gorton and Shaw, 2020?

Thank you for catching this! We have updated the citation to Gorton and Shaw, 2020.

Appendix A1: Keep only information that adds to the manuscript and is not already documented elsewhere: location, measurement method, campaign duration, data completeness, uncertainties, QC, and post-processing.

We have removed the diagram and table that detail the instrumentation from Appendix A1, as this information can be found elsewhere. Instead, we point the reader to Severy et al. (2021).

Lines 541-542: "A detailed discussion of the instrumentation aboard the buoy, which did not change between the California and Hawai'i deployments, is provided in Severy et al. (2021)."

Appendices A2-A3 are specific to the O'ahu field deployment, data recovery, post-processing, and quality control, which are unpublished elsewhere, so we have left them intact.

*Severy, M., Gorton, A. M., Krishnamurthy, R., and Levin, M. S.: Lidar Buoy Data Dictionary: For the 2020 – 2021 California Deployments, Pacific Northwest National Laboratory (PNNL), Richland, WA, USA, PNNL-30947, https://doi.org/10.2172/1987710, 2021.*

Appendix A3.3 and A3.4: Consider removing if not essential to understanding the manuscript.

Based on your helpful suggestion here, we recognized that we were not utilizing the buoy observations to their full potential to characterize the performance of ERA5 and UH-WRF. As clouds and waves are both important influences on offshore wind profiles, we have added Sections 3.5 (Wind dataset performance according to cloud conditions) and 3.6 (Wind dataset performance according to ocean conditions) to the main body of the manuscript.

Lines 344-360:

**"3.5 Wind dataset performance according to cloud conditions**

The presence of clouds, and the ability of simulation datasets to account for them, can impact the accuracy of wind speed estimates. For example, Lee et al. (2025) find that rotor layer wind speeds at two locations offshore of California, United States are generally stronger under clear sky conditions and that the High-Resolution Rapid Refresh model exhibits a smaller bias under cloudy conditions. During the O'ahu deployment, the lidar buoy was equipped with a pyranometer that allows for estimation of the cloud mask and cloud optical thickness (Appendix A3.4) (DOE, 2025b). A pyranometer outage occurred between 10 January and 17 March 2023 (DOE, 2025b). Of the period when the wind speed and pyranometer observations, along with the ERA5 and UH-WRF simulations, are available, 42% of the analysis period is characterized by clear sky conditions and 58% by cloudy conditions (Figure 12a). When comparing clear versus cloudy periods, we find the opposite results of Lee et al. (2025): the O'ahu 140 m observed wind speeds are generally stronger during cloudy periods (median = 9.75 m s$^{-1}$) than during clear sky periods (median = 8.57 m s$^{-1}$) (Figure 12b). This is primarily due to the presence of stratocumulus cloud conditions over California, which generally suppress turbulence and vertical mixing, reducing wind speeds, in contrast to trade wind cumulus clouds near Hawai'i, which are associated with stronger winds and increased convection. At the O'ahu location, ERA5 performs slightly worse during cloudy conditions in representing the observed 140 m wind speeds (median bias = -1.52 m s$^{-1}$ for clear sky periods versus -1.69 m s$^{-1}$ for cloudy periods), whereas UH-WRF performs similarly regardless of the presence of clouds (median bias = -0.35 m s$^{-1}$ for clear sky periods versus -0.33 m s$^{-1}$ for cloudy periods) (Figure 12c, d)."

[Figure]

**Figure 12.** (a) Estimated cloud optical thickness based on pyranometer measurements from the lidar buoy deployment. (b) Observed 140 m wind speed, (c) ERA5 140 m wind speed bias, and (d) UH-WRF 140 m wind speed bias according to clear versus cloudy sky conditions.

*Lee, J., Ghate, V. P., Mitra, A., Miller, L. M., Krishnamurthy, R., and Egerer, U.: Characterization of HRRR-simulated rotor layer wind speeds and clouds along the coast of California, Wind Energy Science, 10(11), 2755-2769, https://doi.org/10.5194/wes-10-2755-2025, 2025.*

Lines 365-380:

**"3.6 Wind dataset performance according to ocean conditions**

Over bodies of water, wave fields impact surface momentum fluxes and therefore wind speed profiles (Edson et al., 2013). While some exploratory research has been performed to evaluate the role of wind/wave interactions on wind profiles, most predictive models used for wind resource assessment do not predict the wave fields, instead relying on parameterizations to represent their effects (Gaudet et al., 2022; 2024). Wave measurement at the lidar buoy location begins 17 March 2023 and continues uninterrupted through the remainder of the deployment (Appendix A3.3). During the overlapping period of the wind and wave observations and the ERA5 and UH-WRF simulations, significant wave heights between 1 m and 2 m occur most frequently (63% of the period) (Figure 13a). A positive correlation is noted between observed significant wave height and the 140 m wind speed, though the weakest wind speeds occur when the significant wave heights meet or exceed 1 m (Figure 13b).

Regardless of lidar measurement height, no trends in the ERA5 wind speed performance emerge according to significant wave height, with median ERA5 wind speed biases of -1.72 m s$^{-1}$, -1.31 m s$^{-1}$, -1.58 m s$^{-1}$, and -1.27 m s$^{-1}$ at the closest lidar height to the surface (60 m) corresponding to significant wave heights of 0-1 m, 1-2 m, 2-3 m, and 3-4 m (Figure 13c). At all lidar measurement heights, the UH-WRF wind speed biases become increasingly positive with increasing wave height, with median wind speed biases at 60 m of -0.49 m s$^{-1}$, -0.32 m s$^{-1}$, -0.34 m s$^{-1}$, and -0.02 m s$^{-1}$ corresponding to significant wave heights of 0-1 m, 1-2 m, 2-3 m, and 3-4 m (Figure 13d)."

[Figure]

**Figure 13.** (a) Significant wave height measurements from the lidar buoy deployment. (b) Observed 140 m wind speed, (c) median ERA5 wind speed bias from 60-240 m, and (d) median UH-WRF wind speed bias from 60-240 m according to significant wave height.

Gaudet, B. J., García Medina, G., Krishnamurthy, R., Shaw, W. J., Sheridan, L. M., Yang, Z., Newsom, R. K., and Pekour, M.: *Evaluation of Coupled Wind–Wave Model Simulations of Offshore Winds in the Mid-Atlantic Bight Using Lidar-Equipped Buoys, Monthly Weather Review, 150, 1377-1395, https://doi.org/10.1175/MWR-D-21-0166.1, 2022.*

Gaudet, B. J., García Medina, G., Krishnamurthy, R., Sheridan, L. M., Yang, Z., Newsom, R. K., Pekour, M., Gustafson Jr., W. I., and Liu, J.: *Assessing Impacts of Waves on Hub-Height Winds off the U.S. West Coast Using Lidar Buoys and Coupled Modeling Approaches, Pacific Northwest National Laboratory, Richland, WA, United States, PNNL-35856, https://doi.org/10.2172/2337529, 2024.*